# High-fidelity single-frame computational super-resolution using signal-preserving denoising-enabled deconvolution

Fudong Xue[1,5], Lin Yuan [1,5] ✉, Wenting He[1,5], Zuo'ang Xiang[1,2], Jun Ren[3], Chunyan Shan [4], Shunqin Li[2], Min Wang[1,2], Liangyi Chen [3] & Pingyong Xu [1,2] ✉

Computational super-resolution (SR) methods enable nanoscale imaging from single-frame wide-field or spinning-disk confocal images without hardware modifications, yet face limitations: statistical restoration suffers from noise and artifacts, while deep learning methods typically lack generalizability. We introduce 3Snet-CLID, a computational SR method which integrates a hybrid supervised/self-supervised deep learning network for signal-preserving denoising with direct Richardson–Lucy deconvolution. 3Snet-CLID's per-pixel denoising strategy suppresses noise while maintaining signal distribution, mitigating artifacts, and enhancing robustness. The method achieves more than 5-fold resolution improvement on conventional microscopes, revealing diverse structures such as the mitochondrial outer membrane, endoplasmic reticulum, and nuclear pores in live and fixed cells under standard labeling. By overcoming key computational SR bottlenecks, 3Snet-CLID offers denoising capability and an accessible platform for high-fidelity nanoscale live-cell imaging.

Live-cell fluorescence imaging is crucial for resolving how nanoscale assemblies such as cytoskeletal filaments, membrane nanodomains, and multi-protein complexes organize and remodel in real time inside intact cells, which underlie many key physiological and pathological processes. However, in a conventional far-field microscope, Abbe's diffraction limit[1] constrains lateral resolution to roughly $\lambda/(2NA)$ and axial resolution to several hundred nanometers, so many subcellular features on the 10–100 nm scale—including protein clusters, transport vesicles, and cytoskeletal cross-links—cannot be resolved as distinct objects[2].

The emergence of super-resolution (SR) imaging has overcome this barrier, achieving resolutions at the tens-of-nanometer or even single-molecule level while preserving molecular specificity[2,3]. SR methods fall into two broad categories: optical SR[4–8], which uses

engineered excitation or emission light patterns, and computational SR[9–11], which leverages algorithms to reconstruct high-resolution estimates from diffraction-limited images. Classical Richardson–Lucy (RL) deconvolution is one such computational approach[2,12].

Optical SR techniques can routinely reach 20–100 nm resolution but require dedicated and often complex instrumentation, high illumination doses (risking phototoxicity), and specific fluorophores. They also typically trade off field of view and temporal resolution for spatial detail, limiting their practicality for long-term, high-throughput live-cell imaging[2,12]. In contrast, computational SR can be implemented on standard widefield (WF) or confocal microscopes, reduces demands on illumination intensity and labeling specificity, and is therefore well-suited for live-cell and large-scale studies[2,13–15]. Both deep learning (DL)–based SR (DLSR) and RL-based deconvolution SR

[1]State Key Laboratory of Biomacromolecules, Institute of Biophysics, Chinese Academy of Sciences, Beijing, China. [2]College of Life Sciences, University of Chinese Academy of Sciences, Beijing, China. [3]College of Future Technology, Peking University, Beijing, China. [4]National Center for Protein Sciences, School of Life Sciences, Peking University, Beijing, China. [5]These authors contributed equally: Fudong Xue, Lin Yuan, Wenting He. ✉e-mail: yuanlin@ibp.ac.cn; pyxu@ibp.ac.cn

can operate on single diffraction-limited frames, meaning their temporal resolution is effectively limited only by camera acquisition rates. This single-frame capability simplifies integration into existing workflows, is particularly advantageous for capturing rapid cellular dynamics, and—unlike multi-frame SR methods—avoids the phototoxicity, storage overhead, and temporal constraints associated with acquiring long, high-frame-count time series[15,16].

DLSR methods[13,14,17–20], which learns an image-to-image mapping from low- to high-resolution examples, have demonstrated impressive resolution gains and denoising capabilities in fluorescence microscopy, often surpassing classical deconvolution while enabling rapid 2D/3D reconstruction. However, DLSR models require large amounts of high-quality SR data as ground truth (GT), making training labor-intensive and often impractical due to the rapid dynamics in biological specimens[20–22], and the low-to-SR mapping remains inherently ill-posed[14], leading to strong dependence on the specific structures represented in the training set and potential artifacts when generalizing to unseen organelles or sample types. To mitigate these issues, strategies such as more generalizable architectures, larger and more diverse datasets, and multimodal imaging have been proposed[17,23,24], but these solutions themselves increase system and pipeline complexity, and further raise demands on data volume and computational resources.

RL deconvolution–based SR occupies a special position as a model-based, interpretable computational SR technique that directly incorporates the image-formation physics: the algorithm iteratively maximizes the Poisson likelihood of a latent object given a measured image and a point-spread function (PSF), as first formulated by Richardson and Lucy[25,26]. RL deconvolution has been demonstrated to improve resolution in synthetic, noise-free fluorescence images[27,28]. However, classical RL SR with real microscopic images faces several critical challenges: it is highly sensitive to noise and PSF mismatch, so iterative updates tend to amplify high-frequency noise and produce artifacts unless carefully regularized and appropriately stopped[10,29–31]; its convergence is often ill-conditioned near the diffraction cutoff[25,26,31–33]. A notable recent development is sparse deconvolution SR technology, which aims to reduce noise influence by incorporating sparsity and continuity as priori knowledge into the algorithm[27]. This approach has successfully enhanced the resolution of single-frame WF microscopy to 120–150 nm[27]. However, both sparse deconvolution and other RL-based analytical models rely on assumptions about specimen and image properties to enhance resolution[10,27,34–36], with multiple parameters that require manual tuning, making the process time-consuming and heavily reliant on the selected parameter sets[25–27,34].

Altogether, the constraints of current single-frame computational SR approaches—including training-data dependence and generalization issues for DL, and noise-sensitivity and convergence bottlenecks in RL-based SR—still hinder their widespread, quantitative application to live-cell imaging. To address these challenges, we developed a new RL-based computational SR method that transforms conventional live-cell imaging into a routine SR tool for cell biology, without requiring prior knowledge. Our approach introduces a two-step (DL denoising then deconvolution) technique termed clear image deconvolution (CLID) to enhance resolution. First, we developed a per-pixel denoising strategy called the single-pixel-synchronized switching (3S) method, which enhances the image signal-to-noise ratio by a factor of 5.8 while preserving the original pixel-wise intensity distribution. The noise-free images generated by 3S denoising from fixed reversibly switchable fluorescent proteins (RSFP)-labeled cells serve as GT for training a hybrid supervised/self-supervised denoising network (3Snet). The trained 3Snet is then applied to perform joint denoising and deconvolution of single-frame images (3Snet-CLID) from various cellular structures labeled with conventional fluorescent proteins (FPs) or dyes, thereby improving spatial resolution. 3Snet-CLID is a 2D, single-frame SR technique that achieves approximately 60 nm

resolution on conventional WF and spinning-disk (SD) confocal microscopes, substantially enhancing image contrast. It is suitable for imaging samples under 10 μm in depth and works best with confocal spinning-disk and TIRF microscopy modes.

## Results

### Conception of the single-frame computational SR method 3Snet-CLID

This study mainly focusses on addressing key challenges described above in computational SRs, and enabling SR imaging of single-frame images from WF or SD confocal microscopy across diverse subcellular structures. We propose a two-step strategy, 3Snet-CLID, that combines DL denoising with RL deconvolution to achieve single-frame SR: (1) A DL denoising network is applied to a single noisy input image to obtain a clear image; (2) Standard RL deconvolution is then directly performed on this clear image to generate the SR output. An ideal 3Snet-CLID framework must meet three criteria: (1) The denoising network must be highly effective so that noise is not amplified during subsequent RL deconvolution; (2) The denoised image must preserve the original pixel-wise signal distribution to prevent artifacts during RL deconvolution; (3) The network must generalize well to denoise single-frame images of various subcellular structures.

While self-supervised DL denoising methods are commonly used for single frame denoising, we found that existing approaches often introduce artifacts when their outputs are directly fed into RL deconvolution (Supplementary Fig. 1). We hypothesize that although these methods reduce noise effectively, their denoising process alters the pixel-wise signal distribution, leading to a mismatch with the PSF during RL deconvolution and resulting in artifacts. Therefore, we consider effective denoising coupled with signal distribution preservation as two essential conditions for enhancing spatial resolution in RL-based SR.

To address this, we propose first generating high-quality GT denoised images from multi-frame acquisitions in fixed cells, and then using this GT to train a DL denoising network for single-frame image denoising. For the denoised images to be suitable for CLID, both the GT and the trained network must satisfy specific requirements: (1) The GT must be virtually noise-free while preserving the pixel-wise signal distribution, enabling artifact-free resolution enhancement via RL deconvolution; (2) The network trained on this GT must retain strong denoising performance, preserve the signal distribution similarly to the GT, and exhibit high robustness for denoising single-frame images of different subcellular structures. However, obtaining a suitable GT—a clear image with low noise and preserved signal distribution remains difficult because conventional denoising methods typically rely on explicit noise models tailored to specific noise sources (e.g., dark current, readout noise), which are ineffective against the complex, concurrent noise types encountered in real-world imaging (e.g., shot noise). Moreover, many methods utilize information from neighboring pixels (e.g., Hessian or sparse methods[27,37], Gaussian filter[38], Wiener filter[39]) or frequency-domain operations (e.g., Wavelet transforms[40], Low-pass filter[41]), which inevitably alter the original pixel-wise signal distribution and thus hinder subsequent RL deconvolution.

We therefore introduce a per-pixel denoising strategy that operates on individual pixels independently, rather than using information from surrounding pixels. This approach aims to achieve effective noise suppression while strictly maintaining the original pixel-wise intensity distribution, thereby mitigating noise amplification and reconstruction artifacts during iterative RL deconvolution. Leveraging the property that RSFPs respond to illumination modulation while noise does not, we perform single-pixel-level computations on image series corresponding to the "ON" and "OFF" states of the FP. This enables the effective removal of multiple noise types while preserving the original pixel-wise intensity distribution. The resulting denoised images are used to train a hybrid supervised/self-supervised denoising network,

which is designed to retain the characteristics of the per-pixel denoising strategy and be applicable to CLID SR imaging. We name the denoising method single-pixel-synchronized switching (3S) denoising, the corresponding DL network 3Snet, and the overall SR imaging framework combining 3Snet with RL deconvolution 3Snet CLID.

## Per-pixel 3S denoising strategy

The 3S per-pixel denoising strategy is designed to differentiate signal from noise at the level of individual pixels. This method operates without explicit models of noise or background sources. Instead, we classify noise for each pixel based on its temporal characteristics into two distinct modes: a random component that fluctuates from frame to frame, and a fixed-pattern component that remains constant across frames. The denoising process is achieved by treating each pixel individually and applying two specific operations: multi-frame averaging to suppress the random noise component, and a subtraction operation between carefully defined image sets to eliminate the fixed-pattern noise.

We implement this denoising theory by leveraging RSFPs. The controlled on/off switching of RSFPs, triggered by specific light wavelengths, provides a precise pixel-level correspondence between fluorescent (signal-present) and non-fluorescent (signal-absent) states. Both the acquired "ON" and "OFF" image sets contain the mixed noise (random and fixed). Our 3S-denoising technique, proceeds in two steps: first, multi-frame averaging of the "ON" and "OFF" images independently suppresses the random noise in each state; second, the averaged OFF image (AVG(OFF)) is subtracted from the averaged-ON image (AVG(ON)) to suppress the averaged fixed background, thereby simultaneously isolating the true signal and removing both noise components. This approach preserves the original pixel-wise signal distribution essential for optimal deconvolution.

We developed a denoising strategy depicted in Fig. 1a, involving two steps: signal acquisition and noise removal. Initially, Skylan-S (a green RSFP) underwent repeated switching cycles using 405 nm and 488 nm illumination for ON and OFF states, respectively. Multiple frames of ON and OFF images were captured under the same 488 nm illumination. We then conducted noise reduction by averaging multiple frames for both ON and OFF images to eliminate/suppress random noise (Gaussian and signal-related Poisson noises, et al.). Subsequently, we performed AVG(ON) - AVG(OFF) to eliminate/suppress fixed-pattern systematic noise, resulting in the final denoised image, referred to as the clear image.

## Evaluation of 3S-denoising and 3S-CLID on synthetic microtubule and cellular structure

We first assessed the performance of 3S denoising using synthetic microtubules. To mimic real imaging conditions, background images from a blank region under WF illumination were acquired, and both Gaussian and Poisson noise were added to the synthetic microtubule image. The pixel size was set to 65 nm to match that of the Flash 4.0 camera coupled with a 100× objective. The 3S-denoised clear image substantially outperformed images reconstructed through frame averaging (AVG(ON)), pixel-wise median of image sequences (MED(ON)), and median-based difference images (MED(ON−OFF)), exhibiting substantially higher peak signal to noise ratio (pSNR) and structural similarity index (SSIM) (Fig. 1b). These results indicate that 3S denoising generates higher-quality images with reduced noise and improved signal integrity and distribution compared to conventional averaging.

Both optical and software-based upsampling are critical to RL deconvolution. Optical oversampling provides genuine signal information but reduces the SNR per pixel, whereas software upsampling provides a finer computational grid without adding new signal content. This grid refinement enhances the numerical stability of RL deconvolution and reduces artifacts without degrading SNR. Specifically, RL deconvolution produces substantial gain at high frequencies near the optical cutoff, which is inherently unstable and noise-sensitive. According to Cramér–Rao Lower Bound (CRLB) analysis, the CRLB diverges as spatial frequencies approach the diffraction limit, resulting in severe noise amplification and poor convergence in high-frequency regions[31]. A finer sampling grid helps numerically manage the algorithm's pronounced gain near the cutoff, underscoring the utility of software upsampling for stabilization.

In the current study, we combined modest optical oversampling (65 nm pixel size) and software upsampling prior to RL deconvolution. Synthetic structures denoised with the 3S method were upscaled by a factor of 2, and several upsampling methods were evaluated. Lanczos interpolation[42] produced higher pSNR and SSIM in the resulting 3S-CLID images compared to Linear, Fourier, or Bicubic interpolation (Supplementary Fig. 2). Further testing with Lanczos interpolation at 3× and 4× scaling showed that 3× upsampling achieved the highest pSNR and SSIM, outperforming both 2× and 4× factors (Supplementary Fig. 2). Thus, we use 3× Lanczos upsampling for following study.

We then assessed the performance of 3S-CLID on the denoised images. While RL deconvolution of raw, AVG(ON), or MED(ON) images failed to separate 65 nm lines and introduced artifacts, 3S denoising followed by CLID processing achieved 65 nm resolution with improved contrast (Fig. 1b, c). Although MED(ON-OFF) outperformed AVG(ON), it remained inferior to the 3S denoising approach (Fig. 1b, c). Notably, Fig. 1b highlights excellent performance of 3S-CLID: the reconstruction obtained from a strongly noisy measurement is nearly identical to the reconstruction obtained by deconvolving a blurry but noise-free image. Starting from the raw noisy input (pSNR/SSIM = 8.2/0.15), 3S denoising step substantially suppresses noise while preserving the per-pixel signal distribution, resulting in a denoised estimate with pSNR/SSIM = 47.5/0.98. Applying RL deconvolution to this denoised image (i.e., 3S-CLID) yields a super-resolved reconstruction with pSNR/SSIM = 24.5/0.93, which closely matches the result of deconvolution applied to a blurry, noise-free image (pSNR/SSIM = 23.9/0.94). This near-equivalence indicates that 3S effectively recovers an accurate blurred signal estimate under severe noise, enabling the downstream deconvolution to operate in a regime comparable to the ideal noise-free case. Overall, 3S-CLID combines (i) efficient single-pixel-based denoising, (ii) faithful preservation of the signal distribution, and (iii) high-fidelity SR reconstruction.

We then proceeded by labeling actin with RSFP and imaging fixed cells using WF and SD microscopy. The fluorescence signal acquisition was conducted following the procedure outlined in Fig. 1a. In both instances (Fig. 1d, e, WF and Fig. 1f, g, SD), 3S denoising substantially enhanced the SNR and structural continuity. Furthermore, 3S-CLID improved resolution by factors of 3.7 (WF) and 4.9 (SD), respectively, resolving finer structural details of actin bundles and microfilaments (Fig. 1e, g).

## Design of hybrid supervised/self-supervised DL denoising network 3Snet

DLSR suffers from limited robustness and poor generalization across organelles, as it learns an ill-posed, end-to-end mapping where a single low-resolution pixel can correspond to many high-resolution outputs, causing overspecialization. In contrast, DL-based denoising offers greater robustness[43] by enforcing a strict pixel-by-pixel mapping that preserves the original signal distribution. Inspired by 3S-CLID, we propose to leverage the inherent robustness of denoising networks to overcome the generalization limitations of DLSR.

A major obstacle in supervised denoising is obtaining reliable GT data for training. We postulate that per-pixel 3S denoising method, which effectively removes noise while preserving signal fidelity and spatial distribution, is well-suited to produce accurate and reliable GT images. We hypothesize that a denoising network trained on such data will retain these pixel-level preservation properties. The network's output would maintain the original signal distribution, yielding

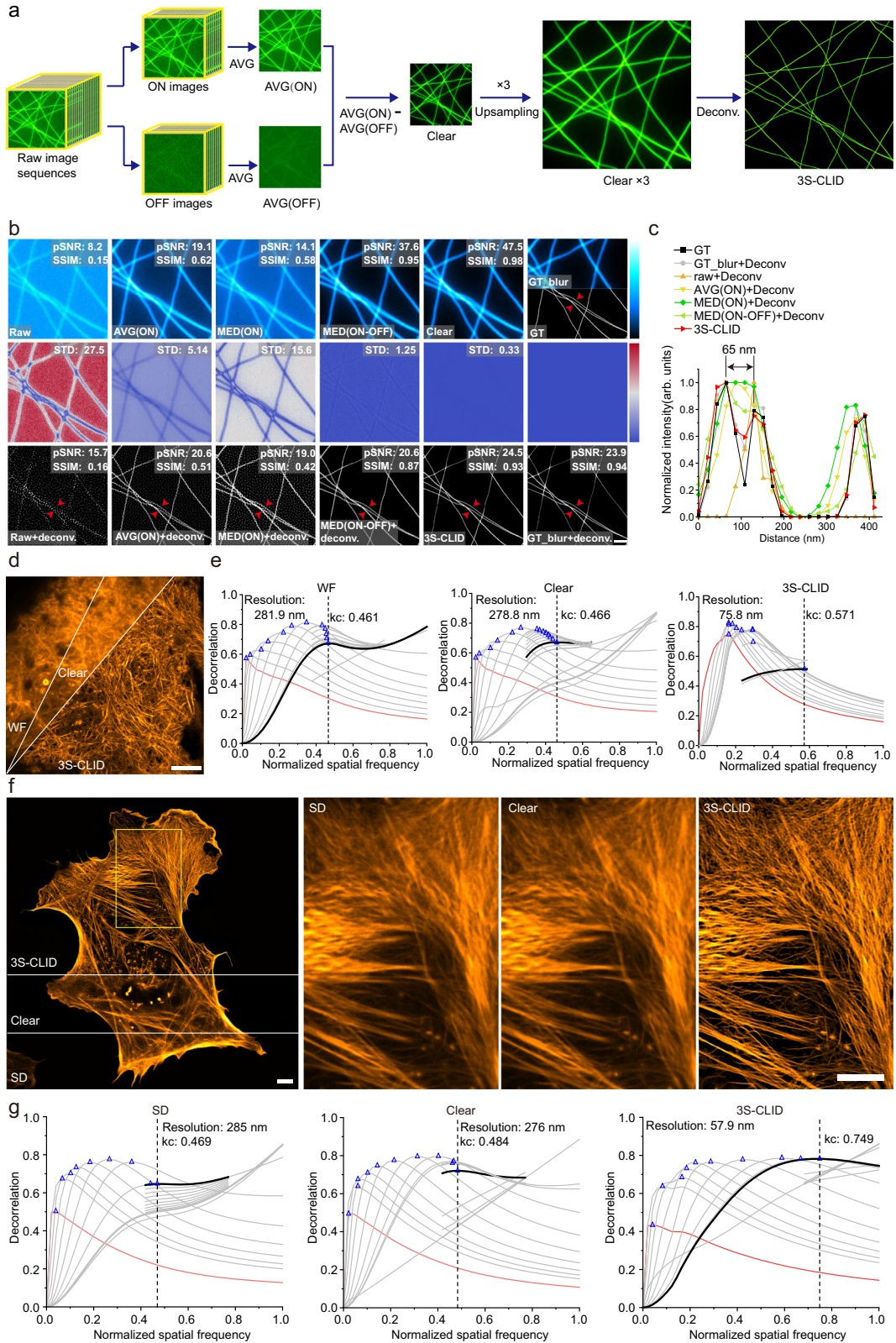

denoised images that are directly applicable to RL deconvolution for enhanced spatial resolution.

Inspired by evidence that incorporating GT improves self-supervised denoising performance[44], we developed 3Snet, a hybrid U-net-based network that integrates supervised and self-supervised learning (Fig. 2a). By averaging ON images with varying frames, we generated noisy images with different noise levels, and randomly selected two noisy images at each noise level to train the self-supervised network for every training iteration. Concurrently, 3S-denoised images were utilized as GT for training supervision (Supplementary Fig. 3). The 3Snet network was employed to denoise single-frame images via WF or SD, subsequently upsampling and applying CLID to accomplish single-frame SR imaging (Fig. 2b). This methodology was termed 3Snet-CLID SR technology.

**Fig. 1 | Workflow and validation of 3S-CLID on synthetic microtubule and cellular structure. a** Overview of 3S-CLID. The workflow involves two sequential steps, 3S-denoising and deconvolved. **b** Validation of 3S-CLID using synthetic microtubule. Top row: peak signal-to-noise ratio (pSNR) and structural similarity (SSIM), from left to right: Raw (noisy image), AVG(ON) (averaging of ON image sequences), MED(ON) (median of ON image sequences), MED(ON-OFF) (median of the ON minus OFF image sequence), Clear (3S-denoising result) and GT/GT_blur (groud truth/blured ground truth). The color bar black to cyan means the relative intensity of images. Middle row: Data uncertainty results of top row indicated by the averaged standard deviation (STD). The color bar, from blue to red, represents the STD value. Bottom row: deconvolution of top row images. The Raw images were created by acquiring a blank region of a fixed cell under WF illumination, and further injection of 5% Poisson noise and 5% Gaussian noise. The resulting images from various approaches were processed using Lanczos upsampling by a factor of

3, followed by RL deconvolution with a theoretically derived Bessel PSF, using 1000 iterations. **c** Normalized intensity profiles along the red arrow heads in the left GT and deconvoled images. **d** 3S-denoised clear and 3S-CLID images of Lifeact-SkylanS-labeled F-actin in fixed U-2 OS cells under wide-field (WF) microscopy. Representative image from four biologically independent replicates is shown. **e** Corresponding decorrelation analysis. The red curve represents the decorrelation function without high-pass filtering; the grey curves show decorrelation functions with high-pass filtering applied; blue triangles indicate local maxima; and the black curve corresponds to the decorrelation function of the highest frequency peak. The vertical dashed line marks the cut-off frequency, kc. Representative data from four biologically independent replicates are shown. **f** 3S-denoised clear and 3S-CLID images of Lifeact-SkylanS-labeled F-actin in fixed U-2 OS cells imaged by Spining-disk (SD) confocal. **g** Decorrelation analysis applied to the SD image for resolution estimation. Scale bars: 5 μm.

## Validation of 3Snet-CLID by standard known structures and cross-validation

We evaluated the performance of 3Snet-CLID in imaging standard structures obtained through WF microscopy. To do this, we conducted a comparison of the outcomes of 3Snet-CLID and direct deconvolution with other denoising methods on an identical image. Initially, we examined fluorescent beads in ball shape of different sizes. Both 3Snet-CLID and sparse deconvolution successfully distinguished the close proximity of 100-nm beads spaced 126-nm apart, whereas Poisson unbiased risk estimate (PURE)[45] and denoising convolutional neural networks (DnCNN)[46] did not achieve the same level of resolution (Fig. 2c). With smaller beads (20 and 40 nm), 3Snet-CLID was able to separate beads spaced 65/66 nm apart—a capability that was not possible with WF microscopy, sparse, PURE and DnCNN deconvolution approaches (Fig. 2c). Following this, we utilized the linear structure of the commercial Argo-SIM as previously reported[27]. Line spacings from 60 nm to 150 nm were indistinguishable under WF microscopy with DnCNN, and Noise2Self[47], but were clearly resolved using 3Snet-CLID (Fig. 2d). Sparse deconvolution can resolve two lines separated by 150 nanometers (Fig. 2d), in agreement with previous reports[27]. PURE and N2V[48] deconvolution introduce significant artifacts into the resulting images (Fig. 2d). For cell samples, 3Snet-CLID more accurately reconstructs endoplasmic reticulum (ER) structures, yielding more continuous, complete morphologies with a higher SNR and less artifacts than other denoising methods (Supplementary Fig. 1). Subsequently, we conducted a cross-validation between 3Snet-CLID and STORM using DNA origami samples labeled with paired fluorophores. The origami structure incorporated two fluorescent sites labeled with FITC-488 molecules, positioned 60 nm apart for STORM microscopy. Although the paired molecules were barely distinguishable under WF microscopy, they were clearly resolved by either 3Snet-CLID or STORM (Fig. 2e). These results emphasize the effectiveness of 3Snet-CLID in processing structures of diverse shapes, leading to a nearly 5-fold enhancement in spatial resolution compared to WF microscopy and almost a twofold improvement compared to sparse deconvolution.

## 3Snet-CLID enable fixed and live-cell single-frame SR imaging

We first assessed the imaging capabilities of 3Snet-CLID on single-frame biological samples from WF and SD imaging. 3Snet-CLID substantially enhanced spatial resolution and contrast in WF imaging, allowing for clear visualization of actin meshes and filaments within a dense actin network in fixed HeLa cells. Additionally, it provided more detailed insight into the actin structures compared to sparse deconvolution (Fig. 3a, b). Using a parameter-free decorrelation analysis method[49], the spatial resolution was improved from 216 nm with WF to 82 nm with 3Snet-CLID reconstruction (Fig. 3c).

Notably, once trained for a specific imaging system, 3Snet-CLID generalizes across different fluorophores without requiring retraining or fine-tuning (Supplementary Fig. 4). Thus, 3Snet-CLID can be effectively utilized in dual-color imaging, improving resolution across

different wavelengths to enhance visualization of fixed cellular structures (Fig. 3d). The dual-color imaging of F-actin (blue) and PMP (peroxisomal membrane protein, magenta) in fixed HeLa cells demonstrated that 3Snet-CLID produced sharper actin filaments with higher contrast compared to WF imaging (Fig. 3d). Additionally, it allowed the previously blurred fluorescent punctum of peroxisomes to be resolved as a ring-shaped structure located within the actin network. In comparison with sparse deconvolution, 3Snet-CLID has an advantage in resolving small ring structures. (Fig. 3d).

In live Jurkat T cells, 3Snet-CLID enabled the analysis of dynamic actin changes during immune synapse formation with single-frame temporal resolution (Fig. 4 and Supplementary Movie 1). The enhanced spatial resolution facilitated detailed observation of the dynamic retrograde flow of lamellipodial actin networks and the centripetal movement of lamellar actin networks corresponding to the distal supramolecular activation cluster (dSMAC) and peripheral supramolecular activation cluster (pSMAC), respectively (Fig. 4a). In line with previous findings[50], the rate of actin retrograde flow across the dSMAC appeared constant, as indicated by the consistent slopes in the kymographs for this region using both 3Snet-CLID and sparse deconvolution techniques (Fig. 4a, dSMAC, red dash lines). However, the centripetal flow rates across the pSMAC were found to be variable, indicated by differing linearities in the kymograph slopes for this area (Fig. 4a, pSMAC, red dash lines), contrary to previous report[50]. Furthermore, both the 3Snet-CLID and sparse deconvolution techniques clearly confirmed that certain actin arcs at the pSMAC converge along the same trajectory over time (Fig. 4a, white arrow heads), highlighting the variability in the velocity of these arcs. Unlike the uniform distribution observed with WF imaging, the enhanced spatial resolution of 3Snet-CLID and sparse deconvolution revealed non-uniform intensity distributions along the trajectories within both dSMAC and pSMAC regions (Fig. 4a). Additionally, variations in the size of actin clusters at the outer edge of the pSMAC were observed at different time points (Fig. 4a), suggesting potential dynamic assembly or disassembly of actin during movement. Moreover, the formation of concentric actin arcs at the boundary between the dSMAC and pSMAC remains ambiguous due to the resolution limitations of WF imaging. In comparison to sparse deconvolution, 3Snet-CLID provides a clearer and more straightforward observation of individual actin bundles or multiple bundles coalescing into an actin arc at the boundary and progressing inward across the pSMAC (Fig. 4a, yellow arrows).

Studies have indicated that actin structures in different regions of the pSMAC layer exhibit diverse orientations and polarities, underscoring the importance of quantifying the anisotropy of these structures to understand their influence on pSMAC actin arc organization[51]. Through the analysis of angles and anisotropy radius using FibrilTool[52], the examination of actin arc anisotropy within the pSMAC region (Fig. 4a, highlighted in yellow and Fig. 4b) demonstrated that, due to enhanced resolution, 3Snet-CLID and sparse deconvolution were more proficient in quantitatively assessing the orientation and polarity of

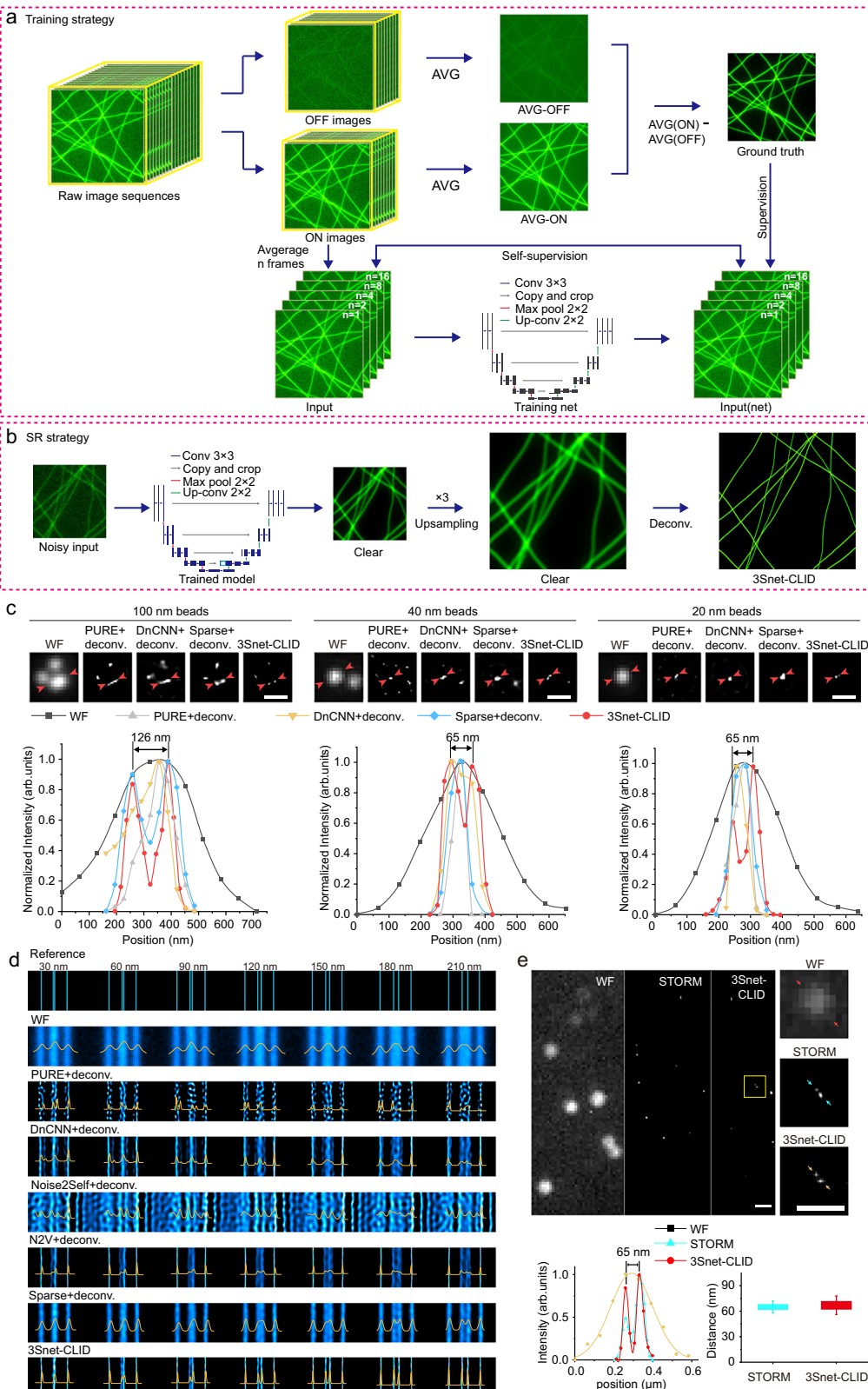

actin arcs compared to WF microscopy. Notably, 3Snet-CLID exhibited the greatest actin arc anisotropy within the pSMAC region in various directions (Fig. 4b). The decorrelation analysis showed the resolution was improved 4.3-fold following 3Snet-CLID processing (Fig. 4c, d).

In a dual-color imaging mode, 3Snet-CLID enabled the visualization of dynamic movement of F-actin, labeled with F-tractin-StayGold (green) and microtubule plus-ends, labeled by EB1-mScarlet-I

(magenta) (Supplementary Fig. 5a and Supplementary Movie 2). Employing a StarDist model provided in TrackMate[53], microtubule plus-ends were automatically tracked over time (Supplementary Fig. 5b). These tracking results demonstrated the ability of 3Snet-CLID to resolve the two poles of the microtubule-organizing center (MTOC) within the cell. Observations revealed the dynamic movement of EB1 proteins radially from the poles along the microtubule length towards

**Fig. 2 | Workflow and validation of 3Snet-CLID using standard structures.**
**a** Training strategy of 3Snet-CLID. The denoised clear images generated by 3S-denoising are utilized as ground truth for training supervision; Noisy images with different noise levels are obtained by averaging ON images with varying frames (1, 2, 4, 8, 16), and then two noisy images at each noise level are randomly selected as a pair to train the self-supervised network for every training iteration. **b** Trained 3Snet-CLID network. A single-frame noisy image is denoised, subsequently upsampled, and applied to accomplish SR imaging. **c** Two adjacent FluoSpheres beads with different sizes were effectively resolved using 3Snet-CLID. Top: Images of 100 nm, 40 nm, and 20 nm beads under WF, PURE+deconvolution, DnCNN+ deconvolution, Sparse+deconvolution, and 3Snet-CLID; Bottom: Intensity profiles along the lines indicated by the arrow heads in Top row head. The peak distances between two adjacent beads with different sizes resolved by 3Snet-CLID are 126, 65,

and 66 nm, while full width at half maximum (FWHMs) of corresponding WF intensity profiles are 310, 254, and 251 nm, respectively. Scale bar: 200 nm. **d** 3Snet-CLID has achieved a spatial resolution of 60 nm on the commercial Argo-SIM under WF microscopy. The intensity profiles of the corresponding double-line pairs (30 nm, 60 nm, 90 nm, 120 nm, 150 nm, 180 nm, and 210 nm distances) are displayed. Representative data from three biologically independent replicates are shown. Scale bar: 500 nm. **e** Cross-validation of 3Snet-CLID and thunderSTORM. Left: DNA origami samples labeled with paired fluorophores of 60 nm distances imaged by WF, thunderSTORM, and 3Snet-CLID; Middle: Zoomed-in regions of yellow box; Right: Intensity profiles along the lines indicated by the arrow heads and the average distance measured using two-peak Gaussian fitting function. Data are presented as mean values ± SD. $n = 7$ from three biological repeats. Scale bar: 500 nm.

the periphery, with varying densities of growing microtubules in different radial directions, indicating a polar generation and distribution of microtubules. Using single-molecule tracking technology, it was observed that the movement trajectories of EB1 molecules mostly did not overlap, with EB1 showing up at different growing microtubule plus ends at distinct time intervals. Interestingly, some EB1 molecules did not stem from the MTOC. As time passed, certain EB1 molecules disappeared earlier within the cell, while others vanished in the distant dSMAC layer of the immune synapse. These results suggest that during the immune synapse formation, EB1 molecules can detach and reattach from specific growing microtubules dynamically, indicating intermittent microtubule growth and dynamic circulation of EB1 proteins along the growing microtubules.

We further demonstrated the SR capability of 3Snet-CLID by applying it to visualize the ring-shaped nuclear pore complex (NPC) labeled with Nup96, a resolution standard for SR microscopy[54]. While conventional SD microscopy depicted Nup96 as diffuse puncta (Fig. 5a), 3Snet-CLID clearly resolved the distinct annular morphology of the NPC (Fig. 5b). The dimensions of the resolved structures (Fig. 5c, d) closely matched those established by electron microscopy[54]. In a final evaluation, we examined the efficacy of dual-color 3Snet-CLID for live cellular structures using SD microscopy. At single-frame temporal resolution, 3Snet-CLID enabled visualization of the dynamic interactions between the ER and mitochondria (Supplementary Movie 3). In the green channel, 3Snet-CLID clearly resolved the outer membranes of closely positioned mitochondria, along with the potentially stretched thin mitochondrial outer membrane (Fig. 5e and Supplementary Fig. 6). In the red channel, 3Snet-CLID also demonstrated diffraction-unlimited spatial resolution, allowing for the precise resolution of finer and sharper tubular ER structures, as well as grid-like tubular ER sheet structures (Fig. 5e and Supplementary Fig. 6). Quantitative assessment showed that the averaged resolution was improved from 363 nm (mitochondria), 346 nm (ER) to 68 nm and 64 nm, respectively, following 3Snet-CLID processing (Fig. 5f). Notably, 3Snet-CLID effectively captured the dynamic changes in the outer membrane structure of mitochondria, a feature that SD cannot analyze, and accurately tracked the dynamic alterations in the tubular and sheet structures of the ER. Using 3Snet-CLID, numerous dynamic events were observed, such as the dynamic generation and disappearance of ER leading to mitochondrial morphology changes, as well as the stretching, recycling, and fusion (Fig. 5g, h). Interestingly, we found previously unreported the lateral fusion event of mitochondrial outer membranes (Fig. 5g, h region 3 and Supplementary Movie 3).

## Discussion

Although RL deconvolution can increase contrast and recover high-frequency content, its practical resolving power is constrained by noise amplification, the ill-posed nature of the inverse problem, and system- and sample-dependent factors such as sampling, pixel-wise intensity statistics, out-of-focus background, and PSF inaccuracies, which together govern the final resolution and structural fidelity.

Building on these considerations, we developed two RL-based super-resolution approaches: 3S-CLID, a multi-frame method for RSFP-labeled fixed cells, and 3Snet-CLID, a DL-enabled single-frame workflow that delivers robust, high-fidelity SR on standard WF and SD confocal microscopes and generalizes to fixed or living samples labeled with conventional FPs or dyes. Both achieve ~60 nm resolution (more than twofold beyond conventional computational SR imaging and >5× beyond the diffraction limit), with 3Snet-CLID providing particularly strong spatiotemporal performance and consistent results across diverse cellular structures.

This performance stems from a tightly coupled set of design choices that jointly improve denoising fidelity and make single-frame deconvolution both stable and high resolving: (1) Per-pixel dual-noise suppression with easy-to-obtain GT. We explicitly separate noise into random and fixed-pattern components based on their spatiotemporal behavior, and use the ON/OFF switching of FPs to isolate the underlying signal at each pixel. This per-pixel synchronization suppresses both noise types while preserving the native intensity statistics, thereby preventing noise-driven or pathological convergence during subsequent RL deconvolution. Importantly, the required GT is obtained in a practical and scalable manner: 3S denoising produces cleaner images with higher signal integrity and a more faithful pixel-wise intensity distribution than conventional averaging (Fig. 1b, c), yielding an ideal training set. (2) Hybrid supervised/self-supervised learning that generalizes to real data. 3Snet integrates supervised learning (enabled by 3S-derived GT) with self-supervised learning (Fig. 2a, b). High-quality supervision anchors the network to physically plausible solutions, while the self-supervised component improves robustness to unknown structures. This hybrid design directly mitigates the generalization limitations frequently observed in purely supervised DL-based SR pipelines. (3) Structure-agnostic denoising that preserves signal statistics critical for resolution. Because 3S/3Snet operates at the single-pixel level, it does not impose priors tied to specific morphology and avoids structural hallucination. We further show that maintaining the original pixel-wise intensity distribution is not merely desirable but essential: it preserves the information needed for accurate deconvolution and prevents artifact formation. In contrast, methods such as DnCNN and N2V can distort native intensity distributions and thereby cap attainable resolution or introduce biases (Fig. 2 and Supplementary Fig. 1). By enforcing distributional integrity, 3Snet-CLID provides a reliable input for deconvolution rather than a visually denoised but statistically altered image. (4) Broad applicability without parameter burden. Although trained on RSFP-based GT, 3Snet effectively denoises images acquired with conventional FPs and dyes, indicating that it learns transferable noise–signal separation rather than probe-specific heuristics. Unlike analytical restoration approaches that often require specimen-dependent assumptions and careful parameter tuning, 3Snet-CLID avoids multiple image-specific parameter sets and enables consistent SR performance with minimal user intervention. Computationally, it is also markedly more efficient, running ~50–240× faster than sparse deconvolution even with optimized

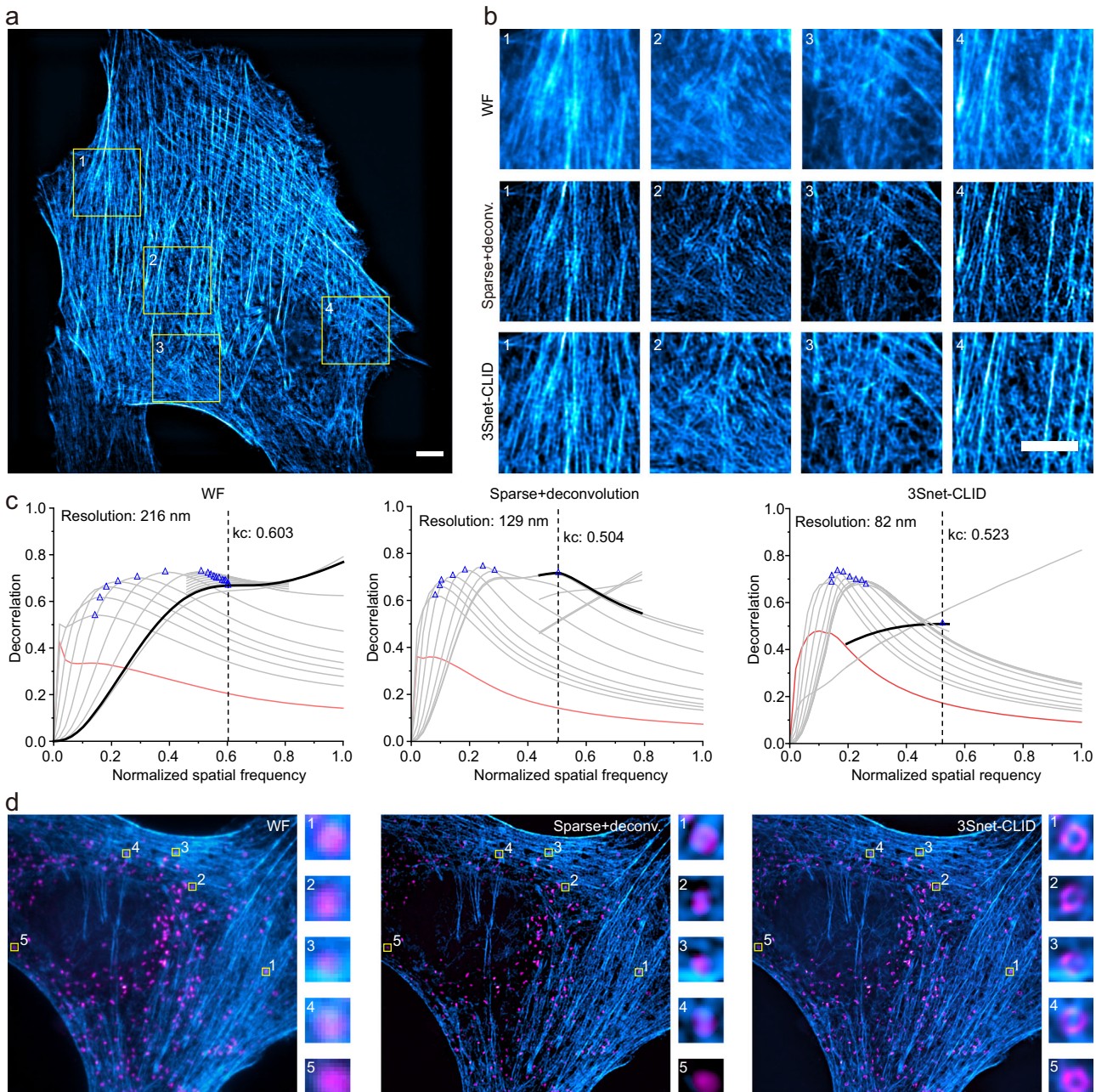

**Fig. 3 | 3Snet-CLID massively improves the quality of cellular structures.**
**a** 3Snet-CLID imaging of F-actin labeled with phalloidin in fixed HeLa cells.
**b** Zoomed-in regions (1–4) from (**a**). Top to Bottom: WF, Sparse+deconvolution, and 3Snet-CLID. **c** Decorrelation analysis applied to the WF images for resolution estimation. **d** Dual-color WF, Sparse+deconvolution, and 3Snet-CLID imaging of F-actin with abberior STAR GREEN-labeled phalloidin (blue) and peroxisomes with abberior STAR RED-labeled PMP70 (magenta) in fixed HeLa cells. Colums: magnified views from white boxed regions. Scale bars: 5 µm (**a**, **b**), 500 nm (**d**).

parameters (Supplementary Table 1). (5) A simplified and robust DL-to-restoration workflow with wide structural coverage. Rather than training directly to SR outputs (which tightly couple performance to microscope-specific SR targets), our pipeline decouples denoising from deconvolution: 3Snet delivers a statistically faithful, high-SNR single-frame image, and resolution enhancement is then achieved via a well-posed deconvolution step. This division simplifies deployment (no SR modules integrated into microscopes) and improves robustness across diverse feature types. Consequently, 3Snet-CLID resolves structures spanning micrometer to nanometer scales—including lines, puncta, rings, tubules, filaments, and near-homogeneous patterns—without structure-specific tuning (Supplementary Table 2 and Supplementary Figs. 7 and 8). Additional synergistic elements, such as

adaptive upsampling and accurate PSF modeling, further enhance stability and fidelity. Collectively, these innovations allow 3Snet-CLID to surpass the practical resolution limits of conventional statistical image restoration under single-frame, low-SNR conditions.

Although 3Snet-CLID substantially improves image quality and suppresses artifacts relative to existing approaches (Figs. 2d, 3b, d and Supplementary Fig. 1), faint texture artifacts may occasionally persist, especially in low-SNR regions (e.g., the central area in Fig. 4a). The resolution of 3Snet-CLID, like that of other computational SR methods, is fundamentally bounded by the Nyquist limit of the pixel sampling and is practically constrained by the SNR. Under our current imaging conditions—6.5 µm camera pixels, 100× objective, and 3× Lanczos upsampling—3Snet-CLID achieves ~60 nm resolution on biological

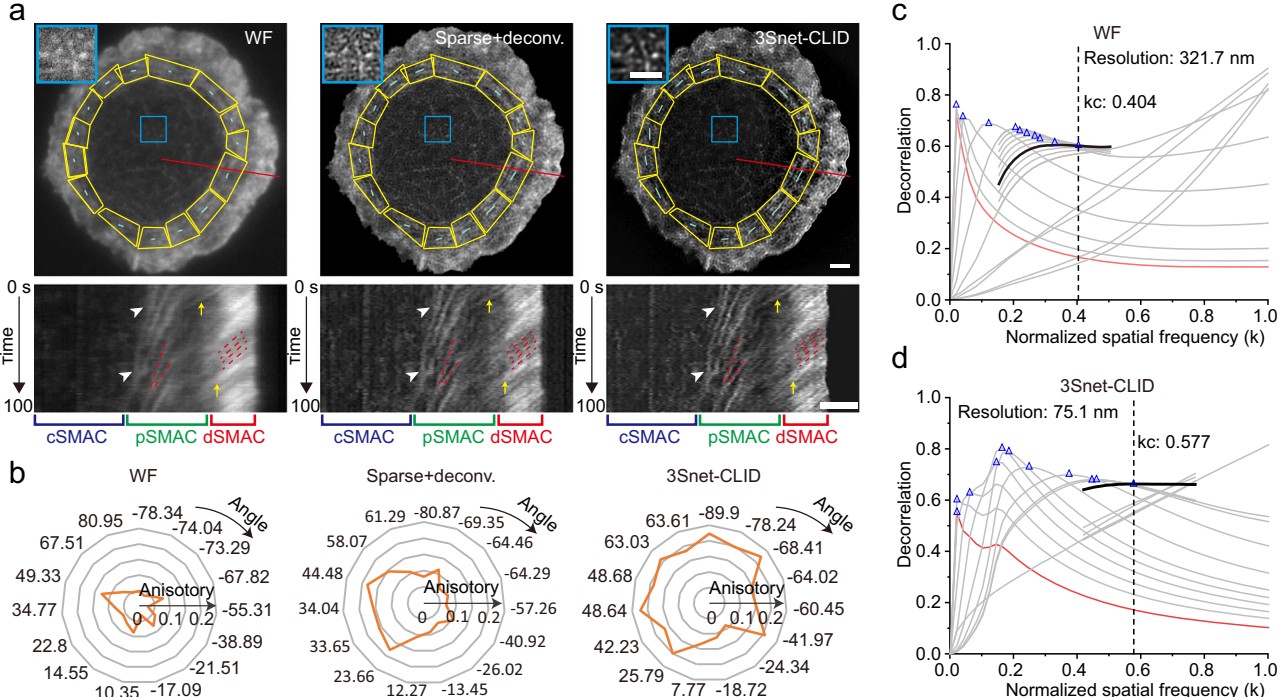

**Fig. 4 | Single-frame live-cell SR imaging with 3Snet-CLID acquired on a WF microscope. a** Utilizing 3Snet-CLID to analyze the intricate actin structures within the immune synapse of Jurkat T cells. Top: WF microscopy, sparse-deconvolution, and 3Snet-CLID imaging of F-actin labeled with F-tractin-StayGold in live Jurkat T cells. Bottom: Kymographs of F-tractin-StayGold in the region corresponding to the red line. Zoomed-in regions were shown in top-left. Representative data from four biologically independent replicates are shown. Scale bar: 2 μm. **b** The actin arcs anisotropy of the peripheral supramolecular activation cluster (pSMAC) in (**a**) measured by FibrilTool. Compared to WF microscopy and sparse-deconvolution, 3Snet-CLID resolves more clear actin arc structure of the pSMAC, and provides more accurate quantified anisotropy data reflecting the effects of actin arc in the pSMAC organization. **c** Decorrelation analysis applied to the WF image for resolution estimation. **d** Decorrelation analysis applied to the 3Snet-CLID image for resolution estimation.

specimens without introducing appreciable artifacts. In the current study, we primarily employ decorrelation analysis to estimate image resolution; however, we acknowledge that this metric has limitations for computational super-resolution outputs. The 2020 Addendum to the original method notes that resolution estimates can be influenced by image-generation or post-processing steps[49], and Hou et al.[55] further demonstrated that deconvolution may yield artificially elevated decorrelation-based resolution largely through high-frequency amplification without a corresponding improvement in fidelity. Thus, decorrelation analysis confirms the presence of high-frequency content but does not independently verify its accuracy. To mitigate this risk, all SR images in our study were visually inspected at optimized iteration numbers to avoid artifacts prior to decorrelation analysis; under these conditions, the resulting resolution estimates closely agreed with FRC-based validation (Supplementary Fig. 9). Fourier Ring Correlation (FRC) analysis of sector test pattern simulations (Supplementary Fig. 7) and of the same cellular structure imaged with two cameras whose native pixel differed (65 nm vs. 110 nm after optical magnification) (Supplementary Fig. 9) confirm that this gain arises from genuine high-frequency signal recovery rather than deconvolution artifacts. In principle, with higher SNR and finer pixel sampling, 3Snet-CLID can approach the corresponding Nyquist limit.

Because RL deconvolution is sensitive to PSF mismatch and can introduce ringing/haloing or spurious fine structures when the forward model is inaccurate, we used condition-matched PSFs and conservative stopping before any visible artifact. Specifically, we constructed a matched PSF from fluorescent-bead measurements acquired on the same system (e.g., FWHM and sampling) combined with known optical parameters (objective/NA, wavelength, pixel size), and we independently validated it by directly measuring the PSF using ultrasmall, ultrabright polymer dots as quasi-point emitters[56]; both PSFs yield nearly indistinguishable RL reconstructions on the same denoised inputs (Supplementary Fig. 10), supporting robustness against reasonable PSF estimation routes when acquired under matched conditions. Although 3Snet-CLID is a 2D SR technique and like any algorithm that uses a 2D PSFs, remains susceptible to out-of-focus blur, it reliably retains SR resolving capability in samples of ~10 μm thickness in both simulation and SD microscopy (Supplementary Fig. 11 and Supplementary Movie 4), and is particularly well suited for internal reflection fluorescence (TIRF) and SD confocal imaging of subcellular structures in adherent cells. While this study focuses on 2D SR, the same denoising–deconvolution strategy is readily extendable to 3D via axial scanning, and developing volumetric modules (3D denoising and 3D depth-/space-variant deconvolution) to improve axial resolution and further suppress out-of-focus blur will be a primary objective of our future work.

## Methods

### 3S-CLID frameworks

We imaged Lifeact-Skylan-S labeled F-actin in fixed U-2 OS cells. The acquisition protocol generated 20 training datasets in under 30 min. Each dataset comprised 50 ON/OFF cycles with a 50-ms exposure per frame. Specifically, Skylan-S was activated with 405 nm illumination (50 ms, 1.29 W/cm²), and multiple ON-state frames were captured under 488 nm illumination (50 ms, 5.39 W/cm²). The same 488 nm illumination was then used to switch the molecules off, followed by an equal number of OFF-state frames captured using identical settings. To obtain a clear image, we averaged the images captured in the ON state and subtracted the average from those taken in the OFF state. The denoised clear image was then upsampled by factor 3 using Lanczos. RL deconvolution was performed on the clear image to enhance its

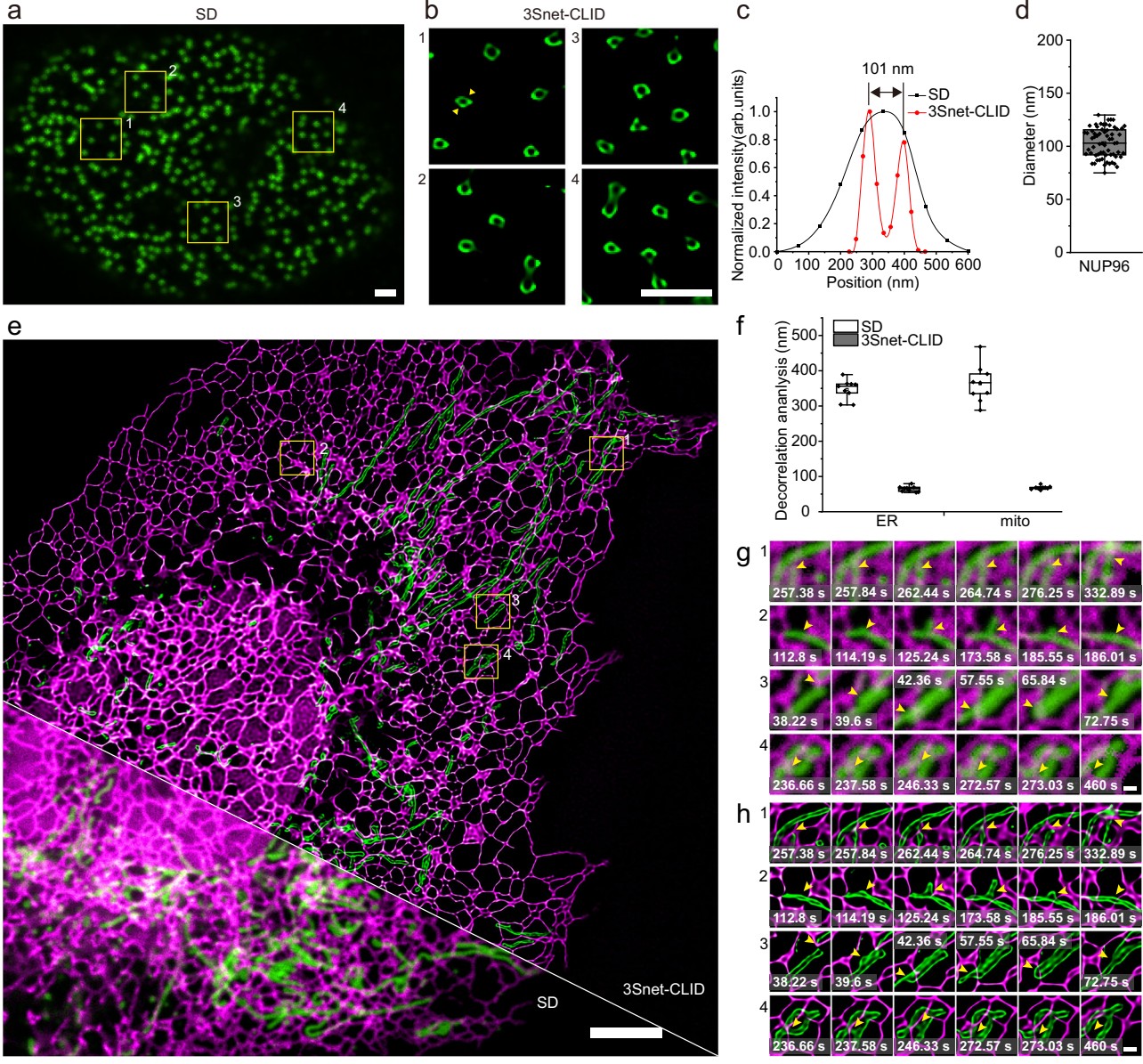

**Fig. 5 | Single-frame SR imaging with 3Snet-CLID acquired on a SD microscope. a** Representative confocal image of the genome-edited fixed U-2 OS cell line with endogenously labeled Nup96-sfGFP. Representative data from from four biologically independent replicates are shown. Scale bar: 2 μm. **b** Zoom in regions (1-4) from (**a**) with 3Snet-CLID reconstruction. Scale bar: 1 μm. **c** Intensity profiles of the nuclear pore indicated by the arrow heads in region 1 of (**b**). **d** Averaged diameters of rings formed by Nup96-sfGFP (103 ± 14 nm; *n* = 64 from three cells). Data are presented as mean ± S.E.M. For the box-and-whisker plots, the center line represents the median, the box spans the interquartile range (IQR; 25th to 75th percentiles), and whiskers extend to 1.5× IQR. Outliers are plotted as individual points.

**e** Dual color imaging of TOM20-StayGold-labeled mitochrondria (green) and mScarlet3-S2-labeled ER (magenta) in live U-2 OS cells. Scale bar: 5 μm. **f** Decorrelation analysis applied to the SD and 3Snet-CLID images for resolution estimation. Data are presented as mean ± S.E.M. For the box-and-whisker plots, the center line represents the median, the box spans the interquartile range (IQR; 25th–75th percentiles), and whiskers extend to 1.5× IQR. Outliers are plotted as individual points. *n* = 9 from three biological replicates. Time-lapse of zoomed-in regions 1–4 from (**e**), showing the corresponding SD (**g**) and 3Snet-CLID reconstruction (**h**) images. Scale bar: 500 nm.

resolution, using Bessel function as the PSF following

$$H = I_0 \left[ \frac{2 * Bessel_1(k \cdot NA \cdot r)}{k \cdot NA \cdot r} \right]^2 \qquad (1)$$

Where $Bessel_1$ is the Bessel function of the first kind of order zero, $I_0$ is the intensity of the incident radiation, $k = 2\pi/\lambda$, $\lambda$ is the wavelength, NA is numerical aperture, $r$ is the circular radius of the polar coordinate. The theoretical 2D PSF used for deconvolution was parameterized based on empirical measurements from fluorescent beads[57].

## Workflow of 3Snet-CLID

The 3Snet-CLID method integrates a three-stage computational imaging pipeline to achieve high-resolution, noise-suppressed images. The process begins with the acquisition of noise-free GT data, uses this data to train a denoising network, and finally applies the network for superior image restoration.

(1).  Data acquisition and GT generation (3S Denoising)

The sample, expressing an RSFP, is imaged under modulated illumination to capture multiple frames in both ON and OFF states. These frames are averaged to produce a mean ON image AVG(ON) and a mean OFF image AVG(OFF). A high-

fidelity, noise-free GT image is then generated by subtracting AVG(OFF) from AVG(ON).

(2). Neural network training

This phase involves training a U-net architecture through two complementary strategies:

i. Self-supervised training:

Images with varying SNRs are synthesized from the acquired data. As shown in Supplementary Fig. 3, to generate training sets with diverse noise levels, we started by gathering 50 frames of images from the ON state and organizing them into five datasets using the combination formula $C_n^k$, where n equals 50 and $k$ is 1, 2, 4, 8, or 16. Next, we randomly selected 50 elements from each dataset and calculated the average of the $k$ images in each chosen element. Thereafter, we randomly selected two images in each dataset as the input ($x$) and target ($y$) images for the U-net model.

ii. Supervised training:

The network is further refined using the high-quality GT image from Stage 1 as a definitive target, enhancing its denoising accuracy.

The resulting predictions were calculated by the following loss function to execute the training stage.

$$loss = \left[ ||\tilde{x} - y||_2^2 + \lambda ||\tilde{x} - GT||_2^2 \right] / (1 + \lambda) \qquad (2)$$

where $\tilde{x}$ represents the network outcome of the input $x$, and $||||_2^2$ refers to the $l_2$ norm, GT is GT. $\lambda > 0$ denotes the weighting factor between the self-supervised term and the supervised term in the loss function. A larger $\lambda$ indicates a higher weight assigned to the supervised term. When $\lambda = 0$, the supervised term becomes zero, and the DL model reduces to a self-supervised model; when $\lambda = \infty$, the self-supervised term becomes zero, and the DL model reduces to a fully supervised model. To determine the optimal $\lambda$, we set $\lambda$ to $0/1/2/4/8/16/\infty$ and trained the model on the same dataset for each setting. The dataset consists of two parts: an image set for training the denoising network (train) and an image set for evaluating the denoising performance (test). The $\lambda$ used in the loss function is chosen as the value that yields the highest average pSNR on the denoised images in the test set.

3) Application: joint denoising and deconvolution

The trained network is deployed to process single-frame images from diverse samples, including those labeled with conventional, photoactivatable, or photoconvertible FPs. A single frame—from either fixed cells or a live-cell time-lapse sequence—is first denoised by the network. The output is then further enhanced by RL deconvolution, resulting in a final SR image.

## Simulations of synthetic microtubule structures

To conduct benchmarks with simulated GT, microtubule-like structures were created using the "Random Walk" process, and the maximal curvatures in the program were utilized[27]. The GT objects were then convolved with a microscope PSF (1.49 NA, 488 nm excitation, 65 nm pixel) to generate simulated blurred ground-truth images (GT_blur). Subsequently, the images were normalized to a 0–255 range and contaminated with Poisson noise (setting lam, the expected number of events occurring, to 5) and Gaussian noise (setting STD, the standard deviation of the distribution, to 5). The raw images were captured by a camera with a pixel size of 65 nm and pixel amplitudes of 16 bits.

## Optical setups

**The WF system.** The WF system was based on a commercial inverted fluorescence microscope (IX83, Olympus) equipped with an objective (100×/NA 1.49 oil, UAPON, Olympus) and laser light with wavelengths of 405 nm (LBX-405-300-CIR-PP, Oxxius), 488 nm (Genesis MX488-500 STM, Coherent), and 561 nm (Jive 500-561 nm, Cobolt). Acoustic optical tunable filters (AOTFnC-400.650-TN, AA Opto-Electronic) were

used to combine, switch, and adjust the illumination power of the lasers. A collimating lens (20×/NA 0.40, Plan N, Olympus) was used to couple the lasers to a polarization-maintaining single-mode fiber (QPMJ-3AF3S, Oz Optics).

The output lasers were collimated and then expanded by a beam expander (GCO-2503, Daheng Optics). The light passed through several mirrors (PF10-03-P01, Thorlabs), and a tube lens (ITL200, Thorlabs) was used to focus on the back focal plane of the objective. A multiband filter cube was used, and emitted fluorescence collected by the same objective passed through a dichroic mirror (ZT405/488/561/640-phase R, Chroma) and an emission filter (FF01-446/523/600/647-25, Semrock). For dual-color imaging, an image splitter (Optosplit II, Cairn) was used before the sCMOS camera (Flash 4.0V2 C11440-22CU, Hamamatsu) to split the emitted fluorescence into two channels.

## The SD confocal microscopy

The Olympus SpinSR10 system utilized an inverted fluorescence microscope (IX83, Olympus) with four lasers (405 nm, 488 nm, 561 nm, and 647 nm) and an objective (100×/NA 1.50 oil, Olympus). The setup included a scanning system (CSU-W1, Yokogawa) and a microscope incubator (Okolab) specifically designed for live cell imaging. The system is operated using CellSens Dimension software, and the image capture is performed using an sCMOS camera (ORCA-Fusion BT, C15440-20UP, Hamamatsu).

The Nikon SoRa system utilized an inverted fluorescence microscope (Ti2-E, Nikon) equipped with four lasers (405 nm, 488 nm, 561 nm, and 647 nm) and a 100×/NA 1.45 oil objective (Nikon). The setup included a scanning system (CSU-W1, Yokogawa) and a microscope incubator (TokaiHit STXG-SP400NX) specifically designed for live-cell imaging. The system was operated using NIS-Elements software, and images were captured with an sCMOS camera (Prime BSI, Photometrics).

## The DeltaVision™ OMX microscope

The DeltaVision OMX system, developed by GE Healthcare, was equipped with an objective (60× PLAPON /1.42 NA), four lasers (405 nm, 488 nm, 561 nm, and 640 nm), four filter sets, and four sCMOS cameras, enabling high-speed imaging in 4-channel modes.

## Standard and cell samples for 3Snet-CLID imaging

**DNA origami materials.** The M13mp18 phage DNA sourced from New England BioLabs was used directly without additional purification. All staple strands, including Alexa Fluor 488-labeled and biotin-modified staple strands, were obtained from Tsingke Biotech (Beijing). The DNA origami staple strands were pre-mixed and stored at -20°C.

The DNA origami nanostructure and synthesis were performed as previously described with slight modifications[58,59]. Briefly, the DNA origami structures were assembled in a single reaction, including 10 nM M13mp18, 100 nM unmodified staple strands, 100 nM biotin-modified strands, and 400 nM Alexa Fluor 488-labeled strands with DNA-PAINT extensions in DNA origami folding buffer. The annealing process, carried out by a PCR device, entails initially holding the temperature at 25 °C for 20 s, then gradually increasing it to 90 °C over 15 min at a rate of 0.5 °C per second. Subsequently, maintaining the temperature at 90 °C for 1 min, and then decreasing it by 1 °C per cycle at a rate of 0.1 °C per second until reaching 70 °C. At 70 °C, maintaining this temperature for 3 min, while decreasing the temperature to 30 °C by 1 °C per cycle at the same rate. After holding it for 30 s, the temperature is finally lowered to 4 °C at a rate of 0.5 °C per second.

For the purification process, a 100 kDa ultrafiltration concentrator was utilized. Initially, 500 μL of DNA origami purification buffer (15 mM MgCl₂ in TAE buffer) was added and centrifuged at 5000 × g at 4 °C for 7 min. Then, 50 μL of annealed solution was mixed with 450 μL of DNA purification buffer and centrifuged at 2000 × g at 4 °C for 20 min. Subsequently, 450 μL of DNA purification buffer was added, and

centrifugation at $2000 \times g$ at 4 °C for 20 min was repeated twice. The filter was then inverted into a fresh tube and centrifuged at $2000 \times g$ at 4 °C for 3 min, and the collected solution was stored at −20 °C.

## STORM imaging

Confocal imaging dishes (Cellvis) were coated with 1 mg/mL biotin-labeled BSA (Solarbio) for 5 min at room temperature. After three washes with PBS, NeutrAvidin (1 mg/mL in PBS) was added to the microwell, incubated for 5 min, and washed three times with PBS. Subsequently, the 60 nm DNA origami structure was diluted in DNA origami imaging buffer (10 mM $MgCl_2$ in PBS) and incubated for 5 min at room temperature, followed by three washes with the same buffer. Once the imager strands were added, the sample was ready for imaging. For the 60 nm structure, an exposure of 200 ms and an illumination power intensity of approximately 4–8 mW/cm² were applied, with 100 frames acquired for the reconstruction. The acquired images were reconstructed with thunderSTORM the Fiji plugin[60].

## Argo-SIM slide

We also utilized a commercial fluorescent sample (the Argo-SIM slide, Argolight) with GT patterns consisting of fluorescing double line pairs (with spacing ranging from 0 nm to 210 nm; $\lambda_{ex} = 300–550$ nm) to validate the resolution (http://argolight.com/products/argo-sim).

## FluoSpheres imaging

To further validate the resolution, FluoSpheres of different sizes (20 nm-, 40 nm-, and 60 nm-diameter, Thermo Fisher Scientific) were diluted with PBS and loaded onto glass coverslips. After allowing the fluorescent beads to settle for 30 min, a single-frame image was taken for analysis.

## Live cell samples

For live 3Snet-CLID experiments, U-2 OS cells or COS-7 cells were labeled with Lifeact-StayGold (Lifeact-SG)/Tom20-SG/Sec61-mScarlet-I using Lipofectamine™ 3000 transfection reagent (Thermo Fisher Scientific) according to the manufacturer's instructions. Jurkat T cells, on the other hand, were transfected with F-tractin-SG/EB1-mScarlet-I via electroporation.

## Fixed cell samples

Cells transfected with genetic indicators were fixed using 4% paraformaldehyde in PBS for 15 min at 37 °C, followed by thorough washing with PBS.

## Cell culture and preparation

The U-2 OS and COS-7 cell lines were purchased from ATCC, while the Jurkat T cell line was acquired from the National Infrastructure of Cell Line Resources. U-2 OS and COS-7 cells were grown in McCoy's 5A medium (Gibco) or DMEM (Gibco), respectively, both supplemented with 10% FBS (Gibco) and 0.1% Mycoplasma prevention reagent (Transgen). Jurkat T cells were cultured in RPMI 1640 medium (Gibco) supplemented with 10% FBS, 1 mM sodium pyruvate solution, 50 μM β-mercaptoethanol (Gibco), and 0.1% Mycoplasma prevention reagent at 37 °C and 5% $CO_2$. For imaging experiments, cells were seeded onto glass coverslips, which were precoated with 10 μg/mL fibronectin for 30 min at 37 °C. For imaging Jurkat T cells, coverslips were coated overnight at 4 °C with 2.5 μg/mL monoclonal anti-CD3 (eBiosciences) and washed with PBS three times. Cell imaging commenced within 10 min after loading cells onto coverslips. To generate homozygous Nup96-sfGFP knock-in U-2 OS cell lines, we employed CRISPR–Cas9-mediated homology-directed repair (HDR). sgRNA sequences[61] were synthesized, annealed, and cloned into the *Bbs*I site of a GFP-deleted pSpCas9(BB)-2A-GFP (PX458) vector (Addgene #48138). Homology arms flanking the gRNA and PAM cleavage sites were amplified and

seamlessly inserted into a pUC19 backbone to generate pUC19-Donor plasmids.

## Electroporation conditions

Cells were seeded at a density between $5 \times 10^5$ and $1 \times 10^6$ cells/mL to ensure that cells were in the exponential growth phase with high viability. Prior to electroporation, the cells were washed with PBS, counted, and resuspended in Opti-MEM™ medium (Gibco) to a cell density of $5 \times 10^6$ cells/mL, and mixed with DNA plasmids (10 μg).

Following gentle mixing of the cells and plasmids in Opti-MEM™ medium (Gibco), the mixture was transferred into electroporation plates with a 0.2 cm gap (Bio-Rad). The electroporation process involved applying a voltage of 120 V, pulse interval of 0.1 s, 2 pulse cycles, and pulse lengths of 50 ms using square waveform pulses (Bio-Rad). Subsequently, 100 μL of the electroporated cells were transferred to 12-well plates containing 1 mL of culture media for Jurkat T cells and were then incubated at 37 °C for 24 h.

## *C. elegans* strains and 3D reconstruction using SD microscopy

The *C. elegans* strain yqIs198 (P$_{hyp-7}$::TOMM-20::GFP) was cultured and maintained under standard conditions[62]. For SD confocal imaging, adult worms were mounted on 2% agarose pads in M9 buffer containing 5 mM levamisole to induce paralysis. For 3D reconstruction, z-series of 100 optical sections was acquired at a step size of 100 nm. Image stacks were processed and analyzed using Fiji (ImageJ).

## Date analysis

We calculated pSNR values between reconstructed images (*I*) and *GT* images by:

$$\text{pSNR} = 10 \log \left( \frac{(peakvalue)^2}{MSE} \right) \qquad (3)$$

$$\text{MSE} = \frac{1}{XY} \sum_{x=1}^{X} \sum_{y=1}^{Y} [GT(x,y), I(x,y)]^2 \qquad (4)$$

Structural similarity (SSIM) values between reconstructed images (*I*) and *GT* images by:

$$\text{SSIM} = \frac{(2\mu_A\mu_B + C_1)(2\sigma_A\sigma_B + C_2)}{(\mu_A^2 + \mu_B^2 + C_1)(\sigma_A^2 + \sigma_B^2 + C_2)} \qquad (5)$$

where $\mu_A$ and $\mu_B$ are the local means, $\sigma_A$ and $\sigma_B$ are the standard deviations for reconstructed images (A) and GT (B) images sequentially.

Standard deviation (STD) is obtained by:

$$\text{STD} = \frac{1}{n-1} \sum_{xy} \sqrt{\left( D(x,y) - \langle D(x,y) \rangle \right)^2} \qquad (6)$$

where *n* is the number of pixels; $D(x, y)$ represents the difference between noisy images and GT images, and $\langle D(x,y) \rangle$ is the averaged intensity of the difference images.

To measure full width at half maximum (FWHM), the intensity profile is fitted by the normal distribution function as following:

$$f(x) = \frac{1}{\sigma\sqrt{2\pi}} \exp[-\frac{(x-x_0)^2}{2\sigma^2}] \qquad (7)$$

where $\sigma$ is the standard deviation and $x_0$ is the expected value, then FWHM is calculated by:

$$\text{FWHM} = 2\sqrt{2\ln(2)}\sigma \qquad (8)$$

The anisotropy of actin arcs was examined using the Fiji plug-in FibrilTool[52]. FibrilTool was integrated into a macro that conducted orientation and polarity analysis within specified regions of interest nested within a grid-like layout that encompassed the overall image. The main fiber orientation was represented as a cyan line in each subregion of interest.

Object tracking was conducted utilizing the TrackMate[53] plug-in for Fiji, and the maximum distance covered by EB1 tracks in each instance in Jurkat T cell was calculated using TrackMate as previously mentioned.

### FRC and the decorrelation analysis

FRC resolution[63] is calculated from two independently recorded images of the same field under identical conditions. For WF, SD confocal, and CLID data, we therefore acquired the sample twice in succession; the resolution is taken as the inverse of the spatial frequency at which the FRC curve falls below the 1/7 threshold. We used the plugin "ImageDecorrelationAnalysis_plugin.jar" to perform the decorrelation analysis[49]. All SR images subjected to decorrelation analysis were first inspected visually at an appropriate iteration number to ensure no obvious texture artifacts were present before the analysis was performed.

### Reporting summary

Further information on research design is available in the Nature Portfolio Reporting Summary linked to this article.

## Data availability

All essential raw datasets, including files for Supplementary Figs. Raw unprocessed images are available at Figshare. Source data are provided with this paper.

## Code availability

The tutorials and the updating version of our 3Snet-CLID software can be found at https://github.com/FudongXue-xpyLab/3Snet-CLID.

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

## Acknowledgements

This project was supported by the National Natural Science Foundation of China (T2394513 P.X., 92254306 P.X., and 32227802 L.Y.), the National Key R&D Program of China (2024YFC3406600 P.X., 2022YFC3400600 L.Y., and 2022ZD0211900 P.X.), and the Strategic Priority Research Program of Chinese Academy of Sciences (XDB37040301 P.X.). We thank Dr. Yuanyuan Li (Institute of Biophysics, Chinese Academy of Sciences) for the help with the DNA PAINT experiment. We thank Dr. Xiaochen Wang (Southern University of Science and Technology) for the gift of *C. elegans* strain. We thank Dr. JunjianWang and Dr. Hongwei Yang (Technical Institute of Physics and Chemistry, Chinese Academy of Sciences) for the gift of the polymer dots.

## Author contributions

P.X. and L.Y. conceived and supervised the study. F.X. performed CLID processing, F.X. and Z.X. analyzed data, L.Y. performed WF and SD imaging, W.H. build the WF microscopy, J.R., and C.S. performed dual-color imaging of PMP and actin in fixed HeLa cells. S.L. generated Nup96-sfGFP knock-in U-2 OS cell line, M.W. cultured the *C. elegans* strain. L.C. participated in project discussion. P.X., and L.Y. wrote the manuscript with contributions from all authors.

## Competing interests

P.X., L.Y., F.X., and W.H. have a pending patent application on the presented framework. The remaining authors declare no competing interests.
