## [Transparent Peer Review file · Nature Communications]

High-fidelity single-frame computational super-resolution using signal-preserving denoising-enabled deconvolution

Corresponding Author: Professor Pingyong Xu

Version 0:

Reviewer comments:

Reviewer #1

(Remarks to the Author)

Review of "Single-frame wide-field and spinning-disk confocal microscopy with 65 nm resolution"

Summary and Assessment

The authors present CLID (Clear Image Deconvolution) microscopy, an innovative super-resolution imaging approach that combines noise removal with deconvolution. This paper introduces two complementary implementations: 3S-CLID, which leverages the optical switching properties of reversibly switchable fluorescent proteins (RSFPs) to separate signal from noise, and 3Snet-CLID, a deep learning-based method that enables single-frame super-resolution imaging.

The authors' approach is highly innovative and commendable. Their method of utilizing ON/OFF states of RSFPs to effectively separate signal and noise components, producing high-quality "clear images," demonstrates excellent understanding of a key challenge in microscopic image processing. Noise reduction prior to deconvolution is crucial for recovering high-resolution components, and this approach significantly enhances the performance of traditional Richardson-Lucy deconvolution. The development of 3Snet-CLID, which uses high-quality images obtained through 3S-CLID as training data for deep learning, further extends the applicability of this technique to single-frame super-resolution imaging. The authors have conducted comprehensive validation using various standard samples, including fluorescent beads, DNA origami, and commercial resolution targets, while also demonstrating excellent performance in live-cell imaging. The computational efficiency, reported to be 50-240 times faster than sparse deconvolution methods, presents a significant advantage for practical applications. This technique is particularly valuable because it can be applied to existing wide-field and spinning-disk confocal microscopes without additional specialized hardware, making it an accessible contribution to the broader research community.

Major Issues

1. Resolution claims could benefit from further contextualization:

Reference (L21-23): "We present CLID (Clear Image Deconvolution) microscopy, powered by deep learning-based denoising, which achieves an unprecedented 65 nm spatial resolution (sixfold improvement) in single-frame images"

The manuscript would benefit from a more detailed explanation of how the 65 nm resolution is achieved in relation to the theoretical limitations of Richardson-Lucy deconvolution when dealing with spatial frequencies near and beyond the diffraction limit. While the authors mention super-resolution techniques in L30-37, providing a clearer definition of "super-resolution" early in the introduction and discussing the theoretical context of the claimed resolution would strengthen the manuscript.

2. The theoretical context could be expanded:

Reference (L59-72): "Deconvolution algorithms and other analytical model-based methods show promise in achieving single-frame SR imaging..."

This section would be enhanced by a more comprehensive discussion of the theoretical considerations in image restoration techniques. Including insights on the challenges faced by statistical image restoration methods when attempting to resolve features beyond twice the diffraction limit (such as those discussed in Liu Y+, Nat Commun 2025) would provide readers with a more complete understanding of the field.

3. Additional perspectives on image upsampling would be valuable:

Reference (L115-117): "To further overcome the Nyquist sampling limit, we upsampled the denoised clear image by a factor of 2 using Lanczos and applied deconvolution (3S-CLID) to enhance the resolution."

The manuscript would be strengthened by discussing the relative merits of software-based upsampling versus optical oversampling during image acquisition. Given that RL deconvolution typically benefits from several-fold oversampling relative to the Nyquist frequency, explaining the rationale behind choosing Lanczos interpolation and addressing how this choice compares to optical oversampling would provide valuable context for readers.

4. The comparative analysis could be more comprehensive:

Reference (L125-129): "The results demonstrated that the clear image produced by the 3S denoising method significantly outperformed the images generated by image averaging (AVG(ON)) and pixel-wise median of image sequences (MED(ON)), with notably higher peak signal-to-noise ratio (pSNR) and structural similarity index (SSIM)"

Including a comparison with MED(ON)-MED(OFF) would provide a more complete evaluation, particularly given that median filtering can be especially effective for Poisson noise reduction. Additionally, providing more detailed information about the RL deconvolution parameters (L344-347) and clarifying whether AVG and MED methods underwent the same 2x upsampling would help readers better understand the experimental conditions.

5. Further validation of generalizability would strengthen the claims:

Reference (L319-324): "Secondly, the denoising network of 3Snet-CLID features a simpler relationship between input and output data, with subsequent deconvolution handled as a separate process. This results in a more robust SR imaging effect that can be applied to a wide array of cellular structures, including lines, puncta, rings, tubules, and filaments, captured by WF and/or SD microscopes"

The manuscript presents promising results for the 3Snet-CLID approach across various cellular structures. This claim would be further strengthened by including additional validation across different cell types, fluorophores, and microscope systems beyond those used for training data acquisition. Such cross-validation would address potential concerns about domain-specific performance that are common in machine learning applications, as noted in L142-148.

Minor Issues

Quantitative characterization of SNR improvement would be beneficial:

Reference (L79-81): "...which leverages reversibly switchable fluorescent proteins (RSFPs) to drastically improve the signal-to-noise ratio (SNR) of the image"

The manuscript would be enhanced by including specific numerical values to quantify the improvement in SNR, rather than using qualitative expressions like "drastically improve."

Figure clarity could be enhanced:

Reference (Fig. 1b): Additional labeling to clearly indicate which methods underwent upsampling prior to deconvolution would improve the interpretability of the figure.

Quantitative assessment of live-cell imaging would add value:

Reference (Fig. 4): Supplementing the impressive visual comparisons in live-cell imaging demonstrations with quantitative metrics of resolution improvement would provide readers with a more objective basis for evaluation.

The connection between learning strategy and Extended Data could be clarified:

Reference (L152-161): "Inspired by the efficacy of supervised-to-self-supervised transfer learning..."

While Extended Data Fig. 2 provides information about dataset generation with varying noise levels for training, the connection between this figure and the self-supervised learning strategy described in the main text could be more explicitly linked. Clarifying how the approach shown in Extended Data Fig. 2 relates to the "supervised-to-self-supervised transfer learning" concept mentioned in L152-161 would help readers better understand this innovative methodology aspect.

(Remarks on code availability)

The necessary information, including the Readme file and installation instructions, is well organized. However, only the 100 nm bead images are attached, and the folder does not include any other images. It would be advisable to upload all images reported in the paper for reproducibility and to enable comparison with other methods.

Reviewer #2

(Remarks to the Author)

In this work by Xue et al., the authors present a new framework, called CLID (Clear Image Deconvolution), for enhancing the resolution of widefield and spinning-disk microscopy images, through a combination of denoising via photoactivation sequences (3S), background removal using activation/deactivation image series, and deconvolution. They further introduce a self-supervised deep learning component, 3Snet, to predict denoised images from single sequences, claiming that this approach achieves 65nm resolution using standard fluorophores and conventional setups.

While the proposed workflow is appealing in its conceptual simplicity and implementation potential, I have significant concerns regarding several of the assumptions, methodological choices, and evaluation strategies used in this study.

A core assumption of the method is that averaging a sequence of photoactivated images yields a noise-free ground truth, suitable for training and deconvolution. However, for dynamic or live samples, such an approach is vulnerable to motion artifacts, particularly motion blur, which could undermine the validity of the averaged image as a clean reference, and pose problem in the deconvolution process. The authors should clarify how they mitigate or compensate for such motion during acquisition, or whether their framework is restricted to fixed or minimally dynamic samples.

In contrast to prior studies, such as Wang et al. 2021, which employ transfer learning to address temporal variability in image sequences, the authors here suggest that a single snapshot suffices to train their denoising model. This assumption should be closely evaluated, especially when considering biological variability across time and space. It remains unclear how a model trained on a single sequence can generalize robustly without overfitting or misrepresenting dynamic features.

The choice of fluorophores also raises practical concerns. The method appears to rely on reversible photoswitchable labels, such as Skyran-S, to distinguish signal from background through the 3S strategy. However, the manuscript does not clarify whether this is a strict requirement. It remains unclear whether the method can operate effectively with conventional fluorophores like GFP, or whether dedicated photoswitchable probes are indispensable. If a photoswitchable fluorophore is indeed required during a preliminary labeling step, the authors should clarify what assumptions are made about their spatial correspondence and background characteristics. Without such clarification, it is difficult to assess the general applicability of the method. These constraints should be explicitly addressed, illustrated with examples if possible, and discussed in terms of feasibility for broad adoption by the biological imaging community.

Similarly, the source and treatment of background signals are not sufficiently explained. The background images acquired during the deactivation phase could include various contributions, such as dark current, autofluorescence, or incomplete photoswitching. A clear explanation of these components is necessary to understand the benefit of this approach over standard background subtraction, particularly under sparse labeling conditions where simple methods often perform well. It is also unclear why the 3S background removal method would outperform conventional strategies unless these sources of signal are well characterized.

The manuscript evaluates its denoising approach mainly in comparison to basic strategies like median or mean filtering. However, recent developments in self-supervised denoising, such as Noise2Void and Noise2Self, have demonstrated strong performance across a range of microscopy contexts. These approaches should be included in the evaluation to contextualize the contribution of 3Snet more rigorously. The authors are encouraged to apply these established methods while keeping the rest of their pipeline consistent, to isolate the impact of the denoising component.

Another important concern relates to the use of 3Snet for denoising images acquired with conventional fluorophores, when the network has been trained on data obtained using reversible photoswitchable fluorophores. This raises questions about the generalization capacity of the model: to what extent can a network trained under specific photophysical conditions reliably predict denoised outputs for fluorophores with different emission characteristics, background profiles, and labeling distributions? The manuscript should explicitly address this important point, and whether retraining or fine-tuning of the network is required for each new fluorophore.

Furthermore, the distinction between averaging images and subtracting dark current images remains ambiguous. The manuscript should explain why 3S denoising outperforms a combination of simple averaging and dark frame subtraction. If the contribution of dark current is dominant in the background signal, then the added value of 3S denoising must be explicitly quantified and justified beyond conventional background approaches much straight forward to implement. For instance, in the simulations presented in Figure 1, a substantial portion of the background could likely be removed through standard dark current correction alone or background subtraction functions. If other sources of background, such as autofluorescence, incomplete fluorophore switching, or camera readout noise, are present, these should be clearly identified, quantified, and their impact on performance thoroughly discussed.

The manuscript lacks details on the acquisition process itself. The authors should report how many images are acquired per

ON/OFF cycle, the acquisition time, and the temporal cost associated with the approach. These factors are particularly important for live-cell imaging, where exposure time, motion, and phototoxicity are limiting factors. Without this information, it is difficult to assess whether the method is broadly applicable in biological settings.

A major technical concern is the use of a theoretical 2D Airy PSF for deconvolution, without any experimental validation and inclusion of 3D blur. Given that spinning-disk and wide-field deconvolution microscopy are often employed for optical sectioning and 3D imaging, the exclusive use of a 2D PSF model appears limiting. Deconvolution in microscopy typically requires a 3D PSF model to handle out-of-focus contributions, especially for thicker specimens, more especially because the main contribution of blur comes from the axial direction. If the method is only valid for flat or 2D structures, this should be explicitly stated. Otherwise, the authors should validate their technique using 3D experimental and simulated datasets to demonstrate broader applicability.

The simulations and reconstructions shown in the manuscript lack essential parameter information. The nature of the PSF, noise levels, and deconvolution settings such as iteration number and regularization terms are not disclosed. This omission hinders reproducibility and makes it difficult to assess whether the results are generalizable or finely tuned to specific conditions.

There are also noticeable artifacts in the deconvolved images, which resemble mismatches between the PSF model and the experimental imaging conditions. These textures are typical when the PSF does not match the sample or when sampling is suboptimal. The authors should explore whether PSF model, oversampling parameters or improved background subtraction prior to deconvolution could mitigate these effects.

Finally, the resolution claims of 65 nm are based from simulations involving idealized flat 1D filament structures or experimental isolated beads, with no tridimensional structure, which are inherently favorable to single frame deconvolution algorithms and are less prone to artifacts. While such examples can serve as useful proof-of-concept demonstrations, they do not provide sufficient evidence to support claims of generalizable high-resolution performance on 3D objects. More complex and heterogeneous structures should be employed. Synthetic test objects that span a wide range of spatial frequencies, such as the sector test pattern (DOI:10.1364/OE.543403), would be particularly informative, as they allow for the simultaneous assessment of resolution recovery in fine-detail regions and artifact generation in coarser areas.

3D blur contributions arising from extended objects imaged with widefield or spinning-disk microscopy should be incorporated into the simulations to more accurately reflect typical imaging conditions. This would help reveal both the strengths and the limitations of the CLID approach when dealing with volumetric samples, where out-of-focus light can significantly degrade image quality and challenge the effectiveness of single frame image deconvolution.

Notably, the reported resolution of 65 nm corresponds approximately to twice the pixel size after 2× oversampling, raising the possibility that this value may reflect sampling limits or the convergence of deconvolution-induced texture rather than effective resolution gain. The authors should investigate whether the resolution estimate stems from true resolution improvement or is merely a byproduct of digital resampling. Varying the degree of upsampling and analyzing its effect on the recovered resolution would help distinguish between these possibilities.

(Remarks on code availability)

Reviewer #3

(Remarks to the Author)

This manuscript describes a method to improve resolution in microscopy images by using a custom, deep-learning based denoising method before a Richardson-Lucy deconvolution. The denoising method uses reversible switchable fluorescent probes to quickly take 'on' and 'off' images where the 'off' images are intended to model background and noise. A UNet is trained to produce denoised images with the target being a 'denoised' image generated from the 'off' images subtracted from the 'on' images.

Denoising before a RL deconvolution would indeed improve the results. There is a long history of the general concept and the manuscript would benefit from doing a better job of explicitly handling this in the introduction.

My main issue with the paper is the theoretical underpinning of the treatment of the noise, which calls into question the ability of the method to be generalized. It also not compared well to other denoising approaches. The biggest contributions to the noise and background are typically the sample itself through Poisson noise, and out of focus sample (even in optically sectioning methods to some degree). Therefore using the 'off' images to remove that doesn't seem to make sense. Also, you can't subtract noise – subtracting two noisy images makes them more noisy. Averaging multiple images is the same as taking a longer exposure. It seems like what is fundamentally happening is the DL network is learning to produce an image that resembles a longer exposure image (with better SNR) that a noisy image. This isn't necessarily a bad thing, but how this process is presented doesn't make sense.

Comparing to a noise-free RL would be nice in the simulated images section.

The math in the Abstract is confusing because the numbers seem to suggest a factor of 4 improvement compared what is stated as 6.

There are very minor grammatical issues.

(Remarks on code availability)

Version 1:

Reviewer comments:

Reviewer #1

(Remarks to the Author)

The authors have responded sincerely to all comments with extensive additional experiments and detailed theoretical explanations. The technical quality of the work is high, and the comprehensive data disclosure demonstrates scientific integrity.

As pointed out in our initial Major Issues 1 and 2, the term "super-resolution" in the manuscript was unclear and misleading. Through the authors' earnest response to comments from all three reviewers, the essential nature of the method has been clarified: CLID is not an optical super-resolution microscopy technique but rather a computational super-resolution image processing method. This clarification of the paper's scope is highly valuable, and we commend the authors' sincere efforts in this regard. However, the title, abstract, and various sections of the main text still contain expressions and discussions that may mislead readers into confusing it with optical super-resolution microscopy. For example, phrases such as "surpasses the theoretical resolution threshold" and "recover lost details" suggest the measurement of new physical information beyond the diffraction limit, when in reality CLID performs computational estimation from existing measurements without hardware modifications.

We would like to suggest that rather than merely making superficial corrections to avoid misunderstandings, reorganizing the manuscript to clearly position it as a paper on "computational super-resolution image estimation/generation" would not only clarify its scope but also make it more compelling. The technical innovations—such as per-pixel denoising strategy, signal distribution preservation, and hybrid supervised/self-supervised learning—are currently scattered throughout the text. By integrating these elements and explicitly presenting the novelty and advantages as an image processing method, the paper's contribution would become clearer and more appealing to the broad readership of Nature Communications.

Specific comments to the authors' revisions/rebuttals follow:

Major Issues 1 & 2: Contextualization and theoretical framework

The authors have provided extensive theoretical clarifications. We suggest this be reflected throughout the manuscript as discussed above.

Major Issue 3: Image upsampling validation

We appreciate the comprehensive validation with systematic testing and complete data regeneration.

Here, the manuscript reads that the synthetic ground truth has 65 nm pixel sampling (Lines 472-480). The validation showed 65 nm resolution achievement. The results can be interpreted in two ways: negatively, as circular validation where the information limit is predetermined; or positively, as evidence that CLID appropriately stops at the data's inherent limit without generating spurious information (avoiding hallucination). Given the importance of this point for interpreting CLID's capabilities, we suggest either more thorough discussion of this issue. More preferably, additional validation approaches if feasible.

Major Issue 4: Comprehensive comparison analysis

We appreciate the addition of MED(ON-OFF) comparison and the detailed parameter specifications as requested. The results show CLID's advantages over alternative methods, which is encouraging.

However, some numerical values in Fig. 1b, 1c are unexpected. For instance, MED(ON) shows STD of 15.6 versus AVG(ON)'s 5.14, and the performance gap between AVG(ON-OFF) and MED(ON-OFF) is larger than typically expected for Poisson noise. We suggest double-checking these values to ensure the comparison's reliability.

Major Issue 5: Demonstration of generalizability

We highly commend the authors' extensive efforts and sincere response. The comprehensive demonstration across diverse conditions (as summarized in Supplementary Table 2) successfully establishes CLID's excellent generalization capability, making its practical utility much clearer. This is a valuable contribution.

Minor Issues 1-5

We appreciate the authors' sincere and appropriate responses to all minor issues.

(Remarks on code availability)

Reviewer #2

(Remarks to the Author)

Overall, the authors have satisfactorily addressed most of the concerns raised in the initial review, and the methodology is now much better explained. I appreciate the additional clarifications and improvements provided in this new version of the manuscript.

However, a few important points still deserve clarification or refinement:

It would be important to clearly state early in the manuscript (e.g., in the Introduction), that the proposed method is essentially two-dimensional and primarily suited for data with high optical sectioning, such as confocal spinning-disk, TIRF microscopy, or other flat samples. Since the approach relies on a 2D model of PSF blur, it does not account for 3D out-of-focus contributions, which inherently limits its applicability to thicker samples compared to standard 3D deblurring approaches. Explicitly stating this from the beginning would help readers understand the intended scope and limitations of the method.

I remain unconvinced by the choice of datasets, both simulated and experimental, used to demonstrate the generalizability of the approach, as they mainly feature 1D filamentous or network structures. I previously suggested, and still recommend, using simulated data that include objects of varying sizes (from micrometer to nanometer scale), such as sector test patterns. For the experimental part, data such as chromatin or lipid membrane staining would better represent more homogeneous 2D surfaces and could highlight that, once trained, the network can handle structures of diverse morphologies. Such examples would also better illustrate the robustness of the method against artefacts in homogeneous regions.

As artefact management is central to this work and to deconvolution in general, it would be useful to explicitly mention that, although artefacts are reduced compared to other algorithms (as illustrated in many comparative examples), residual texture artefacts may still occur, as visible in some figures (e.g., Fig. 4, central region, and Supplementary Video 1). A short comment on this point would guide readers to interpret results with appropriate caution. Such typical textures seem to occur in regions with low spatial details or dominated by out-of-focus signal.

Typical artefacts are also visible in the Argolight slide imaging for all methods tested (Fig. 2d). Even if 3Snet reconstructions outperform other tested methods, sometimes a single line is visible, and sometimes two, independently of the line-pair distances, which is unexpected and would require some explanations. It would be helpful to add a reference line above the figure showing the true pattern (e.g., are the patterns 1-2-1 or 2-2-2 lines?) and to clarify whether the line thickness remains constant across line-pair distances. Another puzzling point is why line-pairs larger than 120 nm are not properly resolved by any algorithms except 3Snet and sparse deconv, and why 3Snet seems to perform better on 60 nm pattern images compared to 210 nm ones.

The poor performance of Noise2Self is somewhat surprising but can likely be explained by the fact that it was designed to denoise very noisy data. This suggests that evaluating the method's performance as a function of SNR, particularly in low-SNR conditions, would provide valuable insight into its limits.

Finally, I remain cautious regarding the reported absolute resolution value of 65 nm (stated directly in the title). This could reflect algorithmic convergence effects (i.e., texture artefacts) rather than true structural resolution. For example, decorrelation analysis of deconvolved images such as those in Fig. 2d might yield similar or even smaller values, making it difficult to discriminate resolution improvement from artefact enhancement. I understand that it is not a trivial task, and including sector test patterns simulations covering a broad range of spatial frequencies could help clarify this point. Unless such analysis is provided, I would recommend moderating or qualifying this claim.

(Remarks on code availability)

Reviewer #3

(Remarks to the Author)

The authors have provided a thorough response to the reviewers. The comparison of the results of this method to deconvolution of a blurry, but noise free image are nearly identical (fig 2b). If this is truly the performance of the method, than this case should be made even more strongly by the authors.

A few critical issues remain - please explain exactly how the decorrelation analysis was done. This should be by comparing FRC between two completely different data sets on the same sample, otherwise the high frequency info could just be a high-resolution, but wrong.

Please also clarify lambda in the loss model.

I share many of the concerns expressed by the other reviewers in the original reviews, in particular the requirement of the PSF to be known exactly as deconvolution goes much beyond the diffraction limit, otherwise artifacts can be produced. This should be discussed more strongly - although I see there is now a figure comparing the result when using mismatched PSFs.

(Remarks on code availability)

Version 2:

Reviewer comments:

Reviewer #1

(Remarks to the Author)

General Assessment

The revised manuscript represents a substantial improvement. The explicit positioning as "computational super-resolution," the removal of the specific resolution figure from the title, and the honest discussion of limitations (2D constraints, PSF sensitivity, residual artifacts, out-of-focus susceptibility) set an appropriate standard for reporting computational SR methods. Aside from one remaining concern discussed below, the revisions adequately address the reviewers' comments.

The authors are particularly commended for conducting additional dual-camera validation experiments—an effort that goes beyond mere textual revision.

Remaining Concern

The use of decorrelation analysis for resolution estimation is understandable and not inappropriate per se. However, given that the authors have now explicitly framed this work as computational super-resolution, the limitations of this metric when applied to computational SR outputs should be clearly discussed.

Decorrelation analysis confirms the presence of high-frequency content but does not independently verify its accuracy. Hou et al. (eLight 2024) demonstrated that deconvolved images can show higher resolution by decorrelation analysis "largely due to simple amplification of the high-frequency part" while lacking fidelity. The 2020 Addendum to the original method (Descloux et al., Nature Methods) also acknowledges that image generation processes can influence resolution estimates.

Suggestion

Part of this concern can be addressed using the authors' own dual-camera dataset. Cross-FRC analysis between the CLID-processed outputs from the two cameras is strongly recommended. Because the two cameras have different pixel sizes, algorithmically generated high-frequency artifacts would be pixel-grid-dependent and thus uncorrelated between the two outputs, whereas genuine structural information would correlate. This provides more rigorous validation than independent decorrelation analysis of each output and requires only re-analysis of existing data.

If the authors' claims are correct, cross-FRC should show correlation up to ~90 nm, corresponding to the resolution of the lower-resolution camera output. This would provide independent confirmation that structural information in this frequency range is consistently recovered. The ~60 nm claim, however, cannot be validated with this dataset. If the authors wish to further validate higher frequencies, a dual-camera configuration with a smaller pixel size (e.g., using a 2× relay lens to achieve ~55 nm pixels) would be one possible approach.

Recommendation Regarding Resolution Claims

Given these validation limitations, stating a specific resolution figure in prominent locations is potentially misleading. The authors' decision to remove the specific number from the title reflects appropriate caution and is commendable. However, the claim of "~60 nm resolution" still appears in the Abstract. Consistent with the reasoning behind the title revision, I recommend removing this figure from the Abstract.

(Remarks on code availability)

The repository was not accessible at the time of review.

Reviewer #2

(Remarks to the Author)

After carefully reviewing the final revised version of the manuscript, I am pleased to note that the authors have thoroughly addressed all the concerns raised during the previous rounds of review. The revisions have substantially strengthened the manuscript, both in terms of methodological clarity and in the discussion of limitations and validation. I appreciate the authors' careful and constructive responses throughout the review process.

Before final publication, I would kindly ask the authors to verify that the newly added supplementary figures mentioned in the point-by-point response to the reviewers are properly included in the merged final document, as they do not seem to appear in the current. In addition, it would be beneficial to include zoomed-in views of selected images to better visualize the reported improvements as well as potential artefact-prone regions. Such magnified views would help readers more clearly

appreciate both the gains and the limitations of the approach.

(Remarks on code availability)

Reviewer #3

(Remarks to the Author)

The authors have satisfactorily responded to my criticisms.

(Remarks on code availability)

POINT-BY-POINT RESPONSE TO EDITOR AND REVIEWER'S COMMENTS

Response to editor:

We sincerely appreciate the opportunity to revise our manuscript and thank all three reviewers for their valuable comments and suggestions, which have helped improve our work. We have carefully revised the manuscript in response to all comments from the reviewers. Below, we address each comment point by point, and all changes in the main manuscript are highlighted in yellow.

Reviewer #1 (Remarks to the Author):

Review of "Single-frame wide-field and spinning-disk confocal microscopy with 65 nm resolution"

Summary and Assessment

The authors present CLID (Clear Image Deconvolution) microscopy, an innovative super-resolution imaging approach that combines noise removal with deconvolution. This paper introduces two complementary implementations: 3S-CLID, which leverages the optical switching properties of reversibly switchable fluorescent proteins (RSFPs) to separate signal from noise, and 3Snet-CLID, a deep learning-based method that enables single-frame super-resolution imaging.

The authors' approach is highly innovative and commendable. Their method of utilizing ON/OFF states of RSFPs to effectively separate signal and noise components, producing high-quality "clear images," demonstrates excellent understanding of a key challenge in microscopic image processing. Noise reduction prior to deconvolution is crucial for recovering high-resolution components, and this approach significantly enhances the performance of traditional Richardson-Lucy deconvolution. The development of 3Snet-CLID, which uses high-quality images obtained through 3S-CLID as training data for deep learning, further extends the applicability of this technique to single-frame super-resolution imaging. The authors have conducted comprehensive validation using various standard samples, including fluorescent beads, DNA origami, and commercial resolution targets, while also demonstrating excellent performance in live-cell imaging. The computational efficiency, reported to be 50-240 times faster than sparse deconvolution methods, presents a significant advantage for practical applications. This technique is particularly valuable because it can be applied to existing wide-field and spinning-disk confocal microscopes without additional specialized hardware, making it an accessible contribution to the broader research community.

Response:

We sincerely appreciate your positive feedback and the valuable time. Your insightful suggestions have significantly strengthened our manuscript, enhancing its accuracy, clarity, and comprehensiveness. Moreover, they have improved the feasibility of our methods for broad

adoption within the biological imaging community.

Major Issues

1. Resolution claims could benefit from further contextualization:

Reference (L21-23): "We present CLID (Clear Image Deconvolution) microscopy, powered by deep learning-based denoising, which achieves an unprecedented 65 nm spatial resolution (sixfold improvement) in single-frame images"

The manuscript would benefit from a more detailed explanation of how the 65 nm resolution is achieved in relation to the theoretical limitations of Richardson-Lucy deconvolution when dealing with spatial frequencies near and beyond the diffraction limit. While the authors mention super-resolution techniques in L30-37, providing a clearer definition of "super-resolution" early in the introduction and discussing the theoretical context of the claimed resolution would strengthen the manuscript.

Response:

We greatly appreciate your valuable suggestions. As recommended, we have provided a clearer definition of "super-resolution" early in the introduction (Line 31-37 in the revised manuscript, highlighted): "However, inherent limitations arise because the imaging system acts as a low-pass filter: the optical transfer function (OTF) restricts the transmission of spatial frequencies. Consequently, high-spatial-frequency components beyond the classical diffraction limit are suppressed or lost, defining the system's diffraction-limited resolution. Super-resolution (SR) imaging aims to recover these lost fine-scale details—representing the high-frequency end of an object's spectrum—from low-resolution measurements."

We have provided a detailed explanation of how the 65 nm resolution was achieved, contextualizing it within the theoretical limitations of Richardson-Lucy (RL) deconvolution. To further clarify our methodology, we have underscored the fundamental distinctions between our denoising approach and existing methods. Specifically, we emphasize that the single-pixel denoising capability of 3Snet enhances network robustness and preserves pixel-wise signal distribution. These properties are pivotal, as they directly contribute to improved spatial resolution and signal fidelity in the subsequent deconvolution process.

These modifications and clarifications have been primarily incorporated into the Results and Discussion sections. The Conception of 3S-CLID (Line 109-145, highlighted) and Conception of 3Snet-CLID (Line 199-224, highlighted) now offer a more detailed account of the core principles underlying our denoising method, its unique advantages over traditional approaches, and its subsequent impact on signal distribution, high-frequency signal restoration via deconvolution, and spatial resolution enhancement. The Discussion section has been revised to include a comprehensive summary and analysis of these points (Line

369-419).

2. The theoretical context could be expanded:

Reference (L59-72): "Deconvolution algorithms and other analytical model-based methods show promise in achieving single-frame SR imaging..."

This section would be enhanced by a more comprehensive discussion of the theoretical considerations in image restoration techniques. Including insights on the challenges faced by statistical image restoration methods when attempting to resolve features beyond twice the diffraction limit (such as those discussed in Liu Y+, Nat Commun 2025) would provide readers with a more complete understanding of the field.

Response:

We sincerely thank you for your insightful comments and valuable suggestions. As recommended, we have incorporated a discussion regarding the challenges confronting current image restoration techniques in this section (Lines 63-76):

"Statistical or frequency-extrapolation-based image restoration methods hold potential for achieving SR imaging from single frames^{13, 19-25}. These approaches rely on mathematical modeling of the image degradation process—commonly comprising blur (modeled via a point spread function, PSF), noise (typically assumed to be Gaussian or Poisson), and sampling artifacts—along with suitable priors to infer high-frequency information beyond the diffraction limit. While such methods can approach the diffraction limit under high signal-to-noise ratio (SNR) and well-characterized degradation models, reliably resolving features beyond twice the diffraction limit remains a fundamentally ill-posed problem. This is due to the acute sensitivity of high-frequency reconstruction to minute noise or errors^{26, 27}, which often leads to significant artifacts and renders the restoration process unstable and critically dependent on strong—and often unverifiable—prior assumptions^{19, 21, 28}. For instance, the iterative Richardson-Lucy (RL) deconvolution^{23, 29}, though effective in enhancing contrast and recovering certain high-frequency components, encounters inherent limitations near and beyond the diffraction limit, primarily stemming from noise amplification and the severely ill-posed nature of the inverse problem at these spatial frequencies³⁰."

3. Additional perspectives on image upsampling would be valuable:

Reference (L115-117): "To further overcome the Nyquist sampling limit, we upscaled the denoised clear image by a factor of 2 using Lanczos and applied deconvolution (3S-CLID) to enhance the resolution."

The manuscript would be strengthened by discussing the relative merits of software-based upsampling versus optical oversampling during image acquisition. Given that RL deconvolution typically benefits from several-fold oversampling relative to the Nyquist

frequency, explaining the rationale behind choosing Lanczos interpolation and addressing how this choice compares to optical oversampling would provide valuable context for readers.

Response:

We sincerely appreciate the insightful comment. We apologize for the omission in the previous version of the manuscript, where we compared various upsampling methods in Extended Data Fig. 1 but did not elaborate on the comparative results or justify our selection of Lanczos interpolation in the main text. As suggested, we have now included a discussion in the revised manuscript (Lines 156–174) addressing the relative advantages of software-based upsampling versus optical oversampling during image acquisition, their respective implications for RL deconvolution-based SR, the rationale behind our choice of Lanczos interpolation, and all CLID-related results presented in the revised article have been regenerated using $3\times$ upsampling, as detailed below:

“Both optical and software-based upsampling play essential roles in RL deconvolution. Optical oversampling provides genuine signal information at the expense of reduced SNR per pixel, whereas software upsampling does not introduce new signal content but supplies a finer computational grid. This grid refinement enhances the numerical stability of RL deconvolution and suppresses artifacts without incurring an SNR penalty. Specifically, RL deconvolution produces substantial gain at high frequencies near the optical cutoff, which is inherently unstable and highly sensitive to noise. The Cramér–Rao Lower Bound (CRLB) analysis of RL deconvolution indicates that the CRLB diverges as spatial frequencies approach the diffraction limit, resulting in severe noise amplification and poor convergence in high-frequency regions³⁰. This underscores the importance of a finer sampling grid to numerically accommodate the algorithm’s pronounced gain near the cutoff—supporting the utility of software upsampling for stabilization.

In the current study, we employed a combination of modest optical oversampling (65 nm pixel size) and software-based upsampling prior to RL deconvolution. We upscaled 3S-denoised synthetic structures by a factor of 2 and evaluated several upsampling methods. The results demonstrated that the Lanczos interpolation⁴¹ approach yielded higher peak SNR (pSNR) and structural similarity index (SSIM) in the resulting 3S-CLID images compared to linear, Fourier, or bicubic interpolation following RL deconvolution (Extended Data Fig. 1). Further upsampling with Lanczos interpolation at factors of 3 and 4 revealed that $3\times$ upsampling achieved superior pSNR and SSIM relative to both $2\times$ and $4\times$ upsampling (Extended Data Fig. 1). Consequently, all CLID-related results presented in this article have been generated using $3\times$ upsampling. Since the underlying structures remain consistent across different upsampling factors, the resolution enhancement observed in 3S-CLID images reflects a genuine improvement in resolution rather than a mere consequence of digital resampling.”

4. The comparative analysis could be more comprehensive:

Reference (L125-129): "The results demonstrated that the clear image produced by the 3S denoising method significantly outperformed the images generated by image averaging (AVG(ON)) and pixel-wise median of image sequences (MED(ON)), with notably higher peak signal-to-noise ratio (pSNR) and structural similarity index (SSIM)"

Including a comparison with MED(ON)-MED(OFF) would provide a more complete evaluation, particularly given that median filtering can be especially effective for Poisson noise reduction. Additionally, providing more detailed information about the RL deconvolution parameters (L344-347) and clarifying whether AVG and MED methods underwent the same 2 \times upsampling would help readers better understand the experimental

conditions.

Response:

We sincerely appreciate your valuable suggestion. In response, we have incorporated a comparison with the MED(ON)-MED(OFF) method in the revised manuscript. Additionally, we have provided more comprehensive details regarding the RL deconvolution parameters and clarified that both the AVG and MED methods were evaluated under the same 3× upsampling conditions (Line 801-803, Fig. 1 Legend).

The MED(ON)-MED(OFF) results, now included in Fig. 1b, 1c, demonstrate that while MED(ON)-MED(OFF) yields better performance than AVG(ON), it does not achieve the same level of improvement as the 3S denoising approach. (Line 188-189)

Fig. 1 | Workflow and validation of 3S-CLID on synthetic microtubule and cellular structure. b, Validation of 3S-CLID using synthetic microtubule. Top row: peak signal-to-noise ratio (pSNR) and structural similarity (SSIM), from left to right: noisy image, image averaging of ON image sequences, median of ON image sequences, median of the ON minus OFF image sequence, 3S-denoising result (Clear) and blurred ground truth. The color bar black to cyan means the relative intensity of images. Middle row: Data uncertainty results of top row indicated by the averaged standard deviation (STD). The color bar blue to red represents the STD value. Bottom row: deconvolution of noisy image, deconvolution of averaged ON image, deconvolution of median averaged ON image, deconvolution of median ON minus OFF image, 3S-CLID result and ground truth. The noisy images were created by acquiring a blank region of a fixed cell under WF illumination, and further injection of 5% Poisson noise and 5% Gaussian noise. The resulting images from various approaches were processed using Lanczos upsampling by a factor of 3, followed by RL deconvolution with a theoretically derived Bessel PSF, using 1000 iterations. c, Normalized intensity profiles along the red arrow heads in the left ground truth (GT) and deconvolved images.

5. Further validation of generalizability would strengthen the claims:

Reference (L319-324): "Secondly, the denoising network of 3Snet-CLID features a simpler relationship between input and output data, with subsequent deconvolution handled as a separate process. This results in a more robust SR imaging effect that can be applied to a wide array of cellular structures, including lines, puncta, rings, tubules, and filaments, captured by WF and/or SD microscopes"

The manuscript presents promising results for the 3Snet-CLID approach across various

cellular structures. This claim would be further strengthened by including additional validation across different cell types, fluorophores, and microscope systems beyond those used for training data acquisition. Such cross-validation would address potential concerns about domain-specific performance that are common in machine learning applications, as noted in L142-148.

Response:

Thank you for your comments and suggestions. In response, to better demonstrate the general applicability of the 3Snet-CLID method, we have applied it to image intracellular structures across various imaging systems. We have also replaced some of the original images with higher-quality examples for clearer demonstration. In the revised manuscript, we have included a Supplementary Table 2 listing the diverse cell types, fluorophores, and microscope systems used — extending beyond those in the training dataset — and have further discussed the robustness of 3Snet-CLID in the Discussion section (Line 391-404).

Minor Issues

1. Quantitative characterization of SNR improvement would be beneficial:

Reference (L79-81): "...which leverages reversibly switchable fluorescent proteins (RSFPs) to drastically improve the signal-to-noise ratio (SNR) of the image"

The manuscript would be enhanced by including specific numerical values to quantify the improvement in SNR, rather than using qualitative expressions like "drastically improve."

Response:

Thank you for your suggestion. We have quantified the improvement in SNR with specific numerical values. The SNR was increased from 8.2 to 47.5, representing a 5.8-fold improvement. (Line 97-98)

2. Figure clarity could be enhanced:

Reference (Fig. 1b): Additional labeling to clearly indicate which methods underwent upsampling prior to deconvolution would improve the interpretability of the figure.

Response:

We apologize for the lack of clarity in the figure legend. As requested, we have updated the legend to explicitly state that, the resulting images from various approaches were processed using Lanczos upsampling by a factor of 3, followed by RL deconvolution with a theoretically derived Bessel PSF, using 1000 iterations. (Line 801-802).

3. Quantitative assessment of live-cell imaging would add value:

Reference (Fig. 4): Supplementing the impressive visual comparisons in live-cell imaging demonstrations with quantitative metrics of resolution improvement would provide readers with a more objective basis for evaluation.

Response:

Thank you for this valuable suggestion. In response, we have incorporated a quantitative metric to complement the visual comparisons in our live-cell imaging demonstrations. Specifically, we implemented an image resolution estimation method based on decorrelation analysis (A. Descloux, et al., Nature Methods, 2019) to provide an objective assessment of resolution improvement. This addition offers readers a more rigorous and quantitative basis for evaluating the performance of our method.

The updated results, now included in Fig. 4c, d and Fig. 5h, demonstrate not only a noticeable visual enhancement but also a measurable improvement in resolution, as quantified by this approach. (Line 316-317, 348-350)

4. The connection between learning strategy and Extended Data could be clarified:

Reference (L152-161): "Inspired by the efficacy of supervised-to-self-supervised transfer learning..."

While Extended Data Fig. 2 provides information about dataset generation with varying noise levels for training, the connection between this figure and the self-supervised learning strategy described in the main text could be more explicitly linked. Clarifying how the approach shown in Extended Data Fig. 2 relates to the "supervised-to-self-supervised transfer learning" concept mentioned in L152-161 would help readers better understand this innovative methodology aspect.

Response:

Thank you for this insightful suggestion. We apologize for the lack of clarity in our original description. To clarify, our approach was inspired by the cited article's key finding — that incorporating ground truth into a self-supervised framework enhances denoising performance, rather than the use of transfer learning per se.

While the cited method employs two separate networks, our model, 3Snet, uses only a single

network and does not involve any form of transfer learning. Specifically, 3Snet integrates supervised and self-supervised learning within one architecture: the supervised component uses ground truth images (3S-denoised clear images) as targets, while the self-supervised counterpart is trained on the same set of raw images degraded with varying signal-to-noise ratios.

In response to your comment, we have revised Extended Data Fig. 2 to more clearly illustrate the integrated supervised and self-supervised learning pathways of 3Snet.

5. Remarks on code availability:

The necessary information, including the Readme file and installation instructions, is well organized.

However, only the 100 nm bead images are attached, and the folder does not include any other

images.

It would be advisable to upload all images reported in the paper for reproducibility and to enable comparison with other methods.

Response:

Thank you for your valuable suggestions. As recommended, we have now uploaded all images referenced in the manuscript: “All essential raw datasets, including files for supplementary figures and raw unprocessed images, are available at figshare via <https://doi.org/10.1038/s41592-024-02294-7>. Source data are provided with this paper.”

Reviewer #2 (Remarks to the Author):

In this work by Xue et al., the authors present a new framework, called CLID (Clear Image Deconvolution), for enhancing the resolution of widefield and spinning-disk microscopy images, through a combination of denoising via photoactivation sequences (3S), background removal using activation/deactivation image series, and deconvolution. They further introduce a self-supervised deep learning component, 3Snet, to predict denoised images from single sequences, claiming that this approach achieves 65nm resolution using standard fluorophores and conventional setups.

Response:

Thank you for your summary of our manuscript. We would like to clarify that 3Snet integrates supervised and self-supervised learning within a U-net architecture.

1. While the proposed workflow is appealing in its conceptual simplicity and implementation potential, I have significant concerns regarding several of the assumptions, methodological choices, and evaluation strategies used in this study. A core assumption of the method is that averaging a sequence of photoactivated images yields a noise-free ground truth, suitable for training and deconvolution. However, for dynamic or live samples, such an approach is vulnerable to motion artifacts, particularly motion blur, which could undermine the validity of the averaged image as a clean reference, and pose problem in the deconvolution process. The authors should clarify how they mitigate or compensate for such motion during acquisition, or whether their framework is restricted to fixed or minimally dynamic samples.

Response:

We apologize for the insufficient clarity regarding the application scenarios of 3S-CLID and 3Snet-CLID in our original manuscript. 3S-CLID is a multi-frame imaging technique designed exclusively for fixed cells labeled with reversibly switchable fluorescent proteins

(RSFPs). In contrast, 3Snet-CLID is a deep learning-based, single-frame method applicable to both fixed and living cells labeled with any FP.

The noise-free ground truth images are generated using 3S denoising on fixed RSFP-labeled samples. These images are then used either for direct deconvolution with 3S-CLID or for training the denoising neural network underlying 3Snet-CLID. For dynamic or live-cell imaging, it is not necessary to produce new ground truth data using 3S denoising. Instead, the pre-trained 3Snet network — established on fixed cells — is applied to denoise single-frame images, which are subsequently enhanced through deconvolution to improve resolution. Since this approach does not require multi-frame acquisition or processing of live samples, it effectively avoids motion artifacts and motion blur, making it particularly suitable for live-cell applications.

We clearly described the application scenarios of 3S-CLID and 3Snet-CLID in the revised manuscript: “Here, we present two novel SR techniques based on RL deconvolution. The first, 3S-CLID, is a multi-frame method designed for fixed cells labeled with RSFPs. The second, 3Snet-CLID, is a cutting-edge, deep learning-based, single-frame technique that achieves robust, high-fidelity DLSR imaging in both WF and SD systems. It is compatible with both fixed and living cells labeled with any FP or dye. Both methods achieve a spatial resolution of 65 nm, more than doubling the performance of conventional restoration techniques and representing a 5.4 folds improvement over the diffraction limit. This marks the highest reported resolution for a method not requiring a dedicated SR module. 3Snet-CLID, in particular, demonstrates exceptional spatiotemporal resolution and robustness across diverse cellular structures.” (Line 365-373)

2. In contrast to prior studies, such as Wang et al. 2021, which employ transfer learning to address temporal variability in image sequences, the authors here suggest that a single snapshot suffices to train their denoising model. This assumption should be closely evaluated, especially when considering biological variability across time and space. It remains unclear how a model trained on a single sequence can generalize robustly without overfitting or misrepresenting dynamic features.

Response:

Similar to the approach described by Wang et al. (2021), our denoising model was trained using multiple image frames rather than individual snapshots. Specifically, we acquired 20 sets of fixed-cell samples, with each sample imaged across 50 “ON” and 50 “OFF” frames. For each sample, the following procedure was applied:

- 1) *The ground truth image was computed as $AVG(ON) - AVG(OFF)$;*
- 2) *From the 50 “ON” frames, five distinct image groups with varying signal-to-noise ratios*

(SNRs) were generated by averaging different number of raw frames, and each group contains 50 images);

- 3) For each SNR group, two frames were randomly selected to form input-target pairs for self-supervised U-net denoising training;
- 4) The ground truth image served as the target for supervised network training.

The description has been included in Methods (Line 450-465). Please see also the Workflow of 3Snet-CLID (Line 440-470) and Extend Data Fig. 2 for detailed information:

3. The choice of fluorophores also raises practical concerns. The method appears to rely on reversible photoswitchable labels, such as Skyran-S, to distinguish signal from background through the 3S strategy. However, the manuscript does not clarify whether this is a strict requirement. It remains unclear whether the method can operate effectively with conventional

fluorophores like GFP, or whether dedicated photoswitchable probes are indispensable. If a photoswitchable fluorophore is indeed required during a preliminary labeling step, the authors should clarify what assumptions are made about their spatial correspondence and background characteristics. Without such clarification, it is difficult to assess the general applicability of the method. These constraints should be explicitly addressed, illustrated with examples if possible, and discussed in terms of feasibility for broad adoption by the biological imaging community.

Response:

Thank you for your comments. RSFP is needed for both 3S-CLID and 3Snet-CLID. For 3S-CLID, the noise-free image produced by 3S denoising is acquired on fixed cells labeled with reversibly switchable fluorescent proteins (RSFPs), and is used for direct deconvolution (3S-CLID) to improve resolution.

For 3Snet-CLID, noise-free ground truth is produced with RSFP-labeled fixed sample for training 3Snet denoising neural network underlying 3Snet-CLID. Then, the pre-trained 3Snet network is applied to denoise single-frame images of samples labeled with conventional FP/dyes. Specifically, 3Snet-CLID method operates through a sequence of three principal steps: (1) 3S denoising for ground truth generation: we use RSFP under modulated illumination to acquire noise-free ground truth images. (2) neural network training: these denoised images serve as the ground truth for training a u-net-based denoising network. (3) joint denoising and deconvolution: the trained network is applied to denoise single-frame images from samples labeled with a wide range of fluorescent proteins (conventional, photoactivatable, or photoconvertible), which are then further enhanced by RL deconvolution to improve resolution.

Using RSFPs in CLID is straightforward. They share key advantages with conventional FPs: easy fusion to proteins of interest and proven compatibility with SR techniques like PALM/STORM, SOFI, SRRF, NL-SIM, and RESOLFT. Their switching is also easily controlled on standard microscopes by altering excitation wavelengths, similar to conventional multicolor imaging protocols.

*Acquiring the training dataset is efficient, requiring the imaging of only 20 fixed samples (we used actin) labeled with RSFPs on a widefield or spinning-disk confocal microscope, with a total acquisition time typically under 30 minutes. **Notably, each imaging system requires only a single training session. Once complete, the network can be widely applied to denoise images of samples labeled with conventional FPs under standard operating conditions.** The 3SnetCLID framework enables rapid processing, typically taking about 2 seconds to denoise a 512×512 field of view. Thus, our approach is not only straightforward and time-efficient but also offers broad applicability across diverse biological imaging contexts.*

For clarity, we have included the processing steps for 3Snet-CLID in the revised manuscript. (Line 440-470)

4. Similarly, the source and treatment of background signals are not sufficiently explained. The background images acquired during the deactivation phase could include various contributions, such as dark current, autofluorescence, or incomplete photoswitching. A clear explanation of these components is necessary to understand the benefit of this approach over standard background subtraction, particularly under sparse labeling conditions where simple methods often perform well. It is also unclear why the 3S background removal method would outperform conventional strategies unless these sources of signal are well characterized.

Response:

We thank you for this insightful comment. In the revised manuscript, we have now detailed the theoretical foundation of the 3S denoising and its specific advantages over conventional strategies. While we acknowledge the excellent denoising results achieved by existing methods, our primary goal is resolution enhancement through deconvolution after denoising. We developed the 3S method precisely because we found that images denoised with conventional techniques were often not optimal for direct deconvolution, a crucial distinction we have now emphasized in the revised manuscript as following (Line 119-145):

“We posit that CLID’s effectiveness requires both effective denoising and the precise preservation of the original pixel-wise signal distribution. However, conventional denoising methods often fail to achieve this simultaneously because they typically rely on explicit noise models for specific sources (e.g., dark current, readout noise), which are ineffective against the complex, concurrent noise types found in real-world imaging (e.g., shot noise, out-of-focus fluorescence). Furthermore, these methods use information from surrounding pixels (e.g., Hessian or sparse methods^{22, 36}, Gaussian filter³⁷, Wiener filter³⁸) or frequency-domain operations (e.g., Wavelet transforms³⁹, Low-pass filter⁴⁰), which alters the original signal distribution and hinders subsequent deconvolution.

To overcome these limitations, we propose a per-pixel denoising strategy designed to differentiate signal from noise at the level of individual pixels. This method operates without explicit models of noise or background sources. Instead, we classify noise for each pixel based on its temporal characteristics into two distinct modes: a random component that fluctuates from frame to frame, and a fixed-pattern component that remains constant across frames. The denoising process is achieved by treating each pixel individually and applying two specific operations: multi-frame averaging to suppress the random noise component, and a subtraction operation between carefully defined image sets to eliminate the fixed-pattern noise.

We implement this theory by leveraging RSFPs. The controlled on/off switching of RSFPs, triggered by specific light wavelengths, provides a precise pixel-level correspondence between fluorescent (signal-present) and non-fluorescent (signal-absent) states. Both the acquired 'ON' and 'OFF' image sets contain the mixed noise (random and fixed). Our technique, termed single-pixel-synchronized switching (3S) denoising, proceeds in two steps: first, multi-frame averaging of the 'ON' and 'OFF' images independently suppresses the random noise in each state; second, the operation $AVG(ON)-AVG(OFF)$ is performed to subtract the averaged fixed background, thereby simultaneously isolating the true signal and removing both noise components. This approach preserves the original pixel-wise signal distribution essential for optimal deconvolution.

5. The manuscript evaluates its denoising approach mainly in comparison to basic strategies like median or mean filtering. However, recent developments in self-supervised denoising, such as Noise2Void and Noise2Self, have demonstrated strong performance across a range of microscopy contexts. These approaches should be included in the evaluation to contextualize the contribution of 3Snet more rigorously. The authors are encouraged to apply these established methods while keeping the rest of their pipeline consistent, to isolate the impact of the denoising component.

Response:

Thank you for your suggestion. In the revised manuscript, in addition to comparisons with conventional filtering methods (e.g., median and mean filtering) and supervised deep learning approaches (previously shown using ArgoSIM data in Fig. 2), we included new quantitative and qualitative comparisons against self-supervised methods — such as Noise2Void and Noise2Self— applied to our live-cell imaging data (Fig. 2d, Extended Data Fig. 3). To ensure a fair comparison, all methods were evaluated using identical datasets and consistent post-processing pipelines, thereby isolating the performance contribution of the denoising component itself. The results show that 3S method achieves a more effective denoising effect than self-supervised deep learning networks, yielding the highest spatial resolution and the fewest artifacts when used for RL deconvolution.

We believe these additions will provide a more comprehensive and contextualized assessment of 3SNet's capabilities relative to contemporary denoising strategies.

6. Another important concern relates to the use of 3Snet for denoising images acquired with conventional fluorophores, when the network has been trained on data obtained using reversible photoswitchable fluorophores. This raises questions about the generalization capacity of the model: to what extent can a network trained under specific photophysical conditions reliably predict denoised outputs for fluorophores with different emission characteristics, background profiles, and labeling distributions? The manuscript should explicitly address this important point, and whether retraining or fine-tuning of the network is required for each new fluorophore.

Response:

Thank you for your thoughtful comments. We would like to kindly clarify that, once trained for a specific imaging system, our network generalizes across different fluorophores without requiring retraining or fine-tuning. While the initial training is system-specific — necessary to calibrate the network to the particular optical setup — the resulting model achieves robust super-resolution imaging that is agnostic to the fluorescent protein employed.

This generalization is possible because 3Snet learns a denoising process that is independent of fluorophore type. During training, the RSFP is used solely to generate noise-free ground truth images. The network is trained using fluorescent images — specifically, low signal-to-noise ratio inputs and their corresponding ground truths — enabling it to learn a pixel-level correspondence that is not specific to any particular chromophore.

In this regard, 3Snet behaves similarly to established methods such as DnCNN and Noise2Void, generalizing effectively from one fluorescent protein to others unseen during training. A distinct advantage of our approach, however, is that the denoised images produced by 3Snet are particularly well-suited for subsequent deconvolution to enhance spatial resolution. This is because the training process preserves pixel-level signal distribution, a crucial requirement for high-quality deconvolution. Further details can be found in the Concept of 3Snet-CLID section (Line 198-224).

Our experimental results demonstrate that the denoising performance of our RSFP-trained network generalizes effectively across fluorescent probes with vastly different emission spectra, background profiles, and subcellular labeling distributions. This is evidenced in Fig. 3d, which presents a super-resolution image of a sample labeled with a 647 nm dye, denoised using a model trained exclusively on Skylan-S. Moreover, we demonstrated that 3Snet-CLID trained on Skylan S can also resolve structures labeled with conventional red FPs (Extended Data Fig. 4). Critically, despite significant variations in the background fluorescence (Fig. 5e, The background fluorescence levels of the ER are different in regions close to the nucleus compared to regions far from the nucleus) and the distinct morphological distributions of the endoplasmic reticulum and mitochondria (labeled in different colors) within the field of view,

application of our singular trained network yields high-quality super-resolution images across all structures.

7. Furthermore, the distinction between averaging images and subtracting dark current images remains ambiguous. The manuscript should explain why 3S denoising outperforms a combination of simple averaging and dark frame subtraction. If the contribution of dark current is dominant in the background signal, then the added value of 3S denoising must be explicitly quantified and justified beyond conventional background approaches much straight forward to implement. For instance, in the simulations presented in Figure 1, a substantial portion of the background could likely be removed through standard dark current correction alone or background subtraction functions. If other sources of background, such as autofluorescence, incomplete fluorophore switching, or camera readout noise, are present, these should be clearly identified, quantified, and their impact on performance thoroughly discussed.

Response:

*Thank you for your comments. The real-world images contain a complex mixture of noise sources. A strategy that requires explicit identification and sequential removal of each noise type is inherently inefficient and fails to address unknown or uncharacterized noise components. As stated in Response 4, the core innovation of our denoising method is its ability to extract and preserve the true signal distribution across pixels for robust deconvolution, without requiring a priori knowledge of specific noise types. Instead, we leverage the photophysical properties of RSFPs and the 3S acquisition protocol to remove noise based on its spatiotemporal characteristics (both fixed-pattern and random) at the single-pixel level. Crucially, RSFP optical modulation ensures that illumination conditions are identical for the ON and OFF states at each pixel. The $AVG(ON) - AVG(OFF)$ operation is therefore a **single-pixel-dependent, in-cell subtraction** that effectively removes the actual noise present during imaging. Since dark current noise exhibits both fixed and random patterns, our pixel-specific denoising framework simultaneously mitigates it alongside other noise sources.*

While dark current correction can improve image quality, its effectiveness is limited. The acquisition conditions for dark frames differ from those of light frames, meaning a standard dark frame does not fully represent the camera's noise profile under illumination. Consequently, the operation $AVG(ON) - offset$ is suboptimal. As the results showed below, this approach yields a lower signal-to-noise ratio compared to the ground truth (GT) acquired with our 3S method and is prone to introducing artifacts during deconvolution.

A more fundamental issue with traditional background subtraction is its assumption of uniform noise. In reality, noise levels are pixel-dependent. Subtracting a global background value from signal-free pixels alters the intrinsic statistical distribution of the signal across the sensor, which adversely affects subsequent RL deconvolution (see the results showed below,

AVG(ON)-AVG(bg).

- The manuscript lacks details on the acquisition process itself. The authors should report how many images are acquired per ON/OFF cycle, the acquisition time, and the temporal cost associated with the approach. These factors are particularly important for live-cell imaging, where exposure time, motion, and phototoxicity are limiting factors. Without this information, it is difficult to assess whether the method is broadly applicable in biological settings.

Response:

We appreciate your suggestion. Regarding the acquisition of training data, our protocol acquired 20 full training datasets in under 30 minutes. Each dataset consisted of 50 ON/OFF cycles with a 50 ms exposure per frame. It is important to note that during actual sample imaging, the duration is determined entirely by experimental requirements.

A comprehensive description of the 3Snet workflow, along with detailed imaging experimental

parameters — such as the number of images per ON-OFF cycle, acquisition times, and the overall temporal cost of the approach — has been provided in the Methods section (Line 424-426, 440-470).

9. A major technical concern is the use of a theoretical 2D Airy PSF for deconvolution, without any experimental validation and inclusion of 3D blur. Given that spinning-disk and wide-field deconvolution microscopy are often employed for optical sectioning and 3D imaging, the exclusive use of a 2D PSF model appears limiting. Deconvolution in microscopy typically requires a 3D PSF model to handle out-of-focus contributions, especially for thicker specimens, more especially because the main contribution of blur comes from the axial direction. If the method is only valid for flat or 2D structures, this should be explicitly stated. Otherwise, the authors should validate their technique using 3D experimental and simulated datasets to demonstrate broader applicability.

3D blur contributions arising from extended objects imaged with widefield or spinning-disk microscopy should be incorporated into the simulations to more accurately reflect typical imaging conditions. This would help reveal both the strengths and the limitations of the CLID approach when dealing with volumetric samples, where out-of-focus light can significantly degrade image quality and challenge the effectiveness of single frame image deconvolution.

Response:

Thank you for your comment. We have revised the manuscript to clarify that the theoretical 2D point-spread function (PSF) used for deconvolution was parameterized based on empirical measurements from fluorescent beads (Line 437-438).

We agree entirely that background noise and out-of-focus light are central challenges in deconvolution microscopy that significantly impact performance. As the primary focus of our work is to mitigate noise in RL deconvolution, we introduced a practical denoising step to enhance its resolution and reliability. We recognize that in the field of deconvolution microscopy, out-of-focus blur presents a distinct challenge, which we plan to address in a dedicated future study. Nevertheless, we have now included a new analysis using 3D experimental and simulated data to assess how out-of-focus signals affect 3Snet-CLID and to quantify its practical imaging depth.

Our findings, which are now presented in the Discussion section (Line 410-414), confirm that 3Snet-CLID demonstrates robust performance in practical single-cell imaging. Specifically, simulations show that 3Snet-CLID produces reliable super-resolution images without noticeable artifacts in samples up to 10 microns thick. Experimentally, 3Snet-CLID is particularly well-suited for TIRFM and can also excel when paired with optical sectioning techniques such as spinning-disk (SD) confocal microscopy. This is evidenced by imaging in C.

elegans, where CLID achieves excellent performance at depths of up to 10 μm (Supplementary Fig. 1).

In the Discussion section, we stated: “Although CLID is a 2D SR technique and, like other methods using 2D PSFs, affected by out-of-focus blur, it reliably retains SR resolving capability in samples about 10 μm thick for both WF and SD microscopies (Supplementary Fig. 1, Supplementary Video 4). It is particularly suited for TIRFM and SD confocal microscopy for imaging subcellular structures in adherent cells. While this study focuses on 2D SR, the underlying approach holds significant potential for 3D imaging via axial scanning. By extending the denoising and deconvolution framework of 3Snet-CLID into three dimensions, it could be integrated into 3D deconvolution pipelines to improve resolution and suppress out-of-focus blur. Developing such 3D modules—incorporating volumetric denoising and deconvolution—to enhance axial resolution and enable robust volumetric imaging will be a primary objective of our future work.”

10. The simulations and reconstructions shown in the manuscript lack essential parameter information. The nature of the PSF, noise levels, and deconvolution settings such as iteration number and regularization terms are not disclosed. This omission hinders reproducibility and makes it difficult to assess whether the results are generalizable or finely tuned to specific conditions.

Response:

We sincerely apologize for the oversight in not providing the essential parameter details in our original manuscript. In response to this comment, we have now included comprehensive information regarding the PSF model, noise levels, and all relevant deconvolution parameters — such as iteration numbers — in Figure 1 legends. We do not need regularization terms for RL deconvolution when using CLID. This addition will ensure full reproducibility of our simulations and reconstructions and allow readers to better evaluate the robustness and generalizability of our results.

11. There are also noticeable artifacts in the deconvolved images, which resemble mismatches between the PSF model and the experimental imaging conditions. These textures are typical when the PSF does not match the sample or when sampling is suboptimal. The authors should explore whether PSF model, oversampling parameters or improved background subtraction prior to deconvolution could mitigate these effects.

Response:

Thank you for your comments. We fully agree that the choice of PSF model, oversampling strategy, and background handling significantly influence the quality of deconvolved images.

To address these factors systematically, we conducted a series of comparative analyses. First, we evaluated the performance of Gaussian versus Bessel PSF models within the RL deconvolution framework and observed superior reconstruction quality using the Bessel PSF (shown as below).

We also examined multiple upsampling techniques and determined that the Lanczos method provided the optimal balance between resolution enhancement and artifact suppression (Extended Data Fig. 1). Regarding background subtraction, we optimized several acquisition parameters—including exposure time and the number of ON/OFF image frames used for averaging—to minimize noise. A comparative assessment revealed that the AVG(ON) – AVG(OFF) method outperformed simple dark current subtraction or surrounding background subtraction for RL deconvolution in preserving signal integrity (shown as below). Finally, we carefully optimized the number of RL deconvolution iterations to maximize resolution while minimizing the introduction of artifacts. These comprehensive optimizations collectively enhance the robustness and reproducibility of our image processing pipeline.

Although parameter optimization reduces artifacts, residual imperfections persist—a characteristic shared with other RL-based deconvolution techniques. These artifacts likely originate from slight inaccuracies in the PSF model or oversampling parameters, incomplete noise suppression, and interference from defocus signals. To ensure an equitable comparison with state-of-the-art methods (PURE, DnCNN, and Sparse), all approaches were evaluated on the same sample under identical imaging conditions. Our 3Snet method exhibits superior resolution and fewer artifacts compared to these alternatives (Fig. 2d, Extended Data Fig. 3), with comprehensive comparisons detailed in the supplementary material.

Extended Data Fig. 1 | Assessment of upsampling approaches for 3S-CLID. a, 3S-denoised synthetic structures were upsampled by linear, Fourier, Bicubic or Lanczos interpolation approaches for RL deconvolution. 5% Poisson noise and 5% Gaussian noise were added to the acquired a blank region of a fixed cell under WF illumination, and 3S-denoised images were processed using different upsampling approaches by a factor of 2, followed by RL deconvolution with a theoretically derived Bessel PSF, using 1000 iterations. The pSNR and SSIM of 3S-CLID images were measured to compare the performance of the four upsampling approaches. Scale bar: 1 μm . b, 3S-denoised images were processed using Lanczos upsampling approach by different factors, followed by RL deconvolution with a theoretically derived Bessel PSF, using 1000 iterations.

Fig. 2 | Workflow and validation of 3Snet-CLID using standard structures. d, 3Snet-CLID has achieved a spatial resolution of 60 nm on the commercial Argo-SIM under WF microscopy. The intensity profiles of the corresponding double-line pairs (30 nm, 60 nm, 90 nm, 120 nm, 150 nm, 180 nm and 210 nm distances) are displayed.

12. Finally, the resolution claims of 65 nm are based from simulations involving idealized flat 1D filament structures or experimental isolated beads, with no tridimensional structure, which are inherently favorable to single frame deconvolution algorithms and are less prone to artifacts. While such examples can serve as useful proof-of-concept demonstrations, they do not provide sufficient evidence to support claims of generalizable high-resolution performance on 3D objects. More complex and heterogeneous structures should be employed. Synthetic test objects that span a wide range of spatial frequencies, such as the sector test pattern (DOI:10.1364/OE.543403), would be particularly informative, as they allow for the simultaneous assessment of resolution recovery in fine-detail regions and artifact generation in coarser areas.

Response:

We appreciate your comment. We would like to clarify that ArgoSIM structure is three-dimensional, with dimensions of $\phi 0.52 \times 36 \mu\text{m}$, featuring nanometer-scale line spacing that enables nanoscale resolution characterization. In contrast, the referenced "sector test pattern" consists of micrometer-scale features, making it unsuitable for nanometer-resolution

assessment.

In line with established practices, as exemplified in a recent *Nature Methods* publication (2025), we further validated our resolution using biological structures—specifically, nuclear pore complex and actin filaments—together with the *ArgoSIM* standard. We generated homozygous *Nup96-sfGFP* knock-in U-2 OS cell line and evaluated 3Snet-CLID's performance. Results demonstrated that 3Snet-CLID clearly resolved the distinct annular morphology of the NPC (Fig. 5a, 5b). The dimensions of the resolved structures (Fig. 5c, d) closely matched those established by EM (PMID: 31562488).

By applying the full width at half maximum (FWHM) criterion, a widely accepted metric in the field (as supported by PMID: 18796604, PMID: 31501551, PMID: 34782739), we resolved individual actin filaments measuring ~ 60 nm in width at representative locations (shown as below). Moreover, using a parameter-free decorrelation analysis method (PMID: 31451766), the spatial resolution was improved from 285 nm with SD to 58 nm with 3Snet-CLID reconstruction (Fig. 1f, g).

13. Notably, the reported resolution of 65 nm corresponds approximately to twice the pixel size after $2\times$ oversampling, raising the possibility that this value may reflect sampling limits or the convergence of deconvolution-induced texture rather than effective resolution gain. The authors should investigate whether the resolution estimate stems from true resolution improvement or is merely a byproduct of digital resampling. Varying the degree of upsampling and analyzing its effect on the recovered resolution would help distinguish between these possibilities.

Response:

Thank you for your comments. As shown in Fig. 2 of the original manuscript, all methods compared with 3S denoising—including PURE, DnCNN, and Sparse—utilized $2\times$ upsampling, yet only 3S achieved a resolution of 65 nm. As you suggested, we performed additional experiments using $3\times$ and $4\times$ upsampling; however, no further improvement in resolution was observed (Extended Data Fig. 1). These results confirm that the enhanced resolution arises from genuine physical or algorithmic improvements in the imaging capability, rather than from digital resampling artifacts.

Although the resolution was not further improved, upsampling with Lanczos interpolation revealed that $3\times$ upsampling achieved superior pSNR and SSIM relative to both $2\times$ and $4\times$

upsampling (Extended Data Fig. 1). Consequently, all CLID-related results presented in the revised manuscript have been generated using $3\times$ upsampling. This result has been incorporated into the revised manuscript (Line 167-174).

Reviewer #3 (Remarks to the Author):

This manuscript describes a method to improve resolution in microscopy images by using a custom, deep-learning based denoising method before a Richardson-Lucy deconvolution. The denoising method uses reversible switchable fluorescent probes to quickly take ‘on’ and ‘off’ images where the ‘off’ images are intended to model background and noise. A UNet is trained to produce denoised images with the target being a ‘denoised’ image generated from the ‘off’ images subtracted from the ‘on’ images.

Response:

Thank you for your thoughtful summary of our manuscript. While we generally agree with your overview, we would like to clarify that the "OFF" image serves not only to model background and noise but also to establish a pixel-wise correspondence with the "ON" image. This pairing preserves the spatial distribution of the signal when the "OFF" image is subtracted from the "ON" image—unlike conventional denoising methods (e.g., low-pass/Gaussian filtering, OLID, or eccentricity-based approaches), which redistribute or alter the signal's spatial distribution. Preserving this distribution is critical for subsequent deconvolution.

*Conceptually, our method is single-pixel-dependent and differs from traditional denoising techniques. By leveraging light modulation, we distinguish signal from noise and extract the signal while maintaining its spatial integrity. Furthermore, our denoising process combines two key operations: subtraction and multi-frame averaging. Specifically, we train a UNet to generate denoised images, where the target output is derived by subtracting the **averaged** "OFF" image from the **averaged** "ON" image.*

1. Denoising before a RL deconvolution would indeed improve the results. There is a long history of the general concept and the manuscript would benefit from doing a better job of explicitly handling this in the introduction.

Response:

We sincerely appreciate your valuable suggestions. In the Introduction section of the revised manuscript, we have now included an introduction to denoising's role in improving image resolution and quality before RL deconvolution. (Line 63-76)

2. My main issue with the paper is the theoretical underpinning of the treatment of the noise, which calls into question the ability of the method to be generalized. It also not compared well to other denoising approaches. The biggest contributions to the noise and background are typically the sample itself through Poisson noise, and out of focus sample (even in optically sectioning methods to some degree). Therefore using the 'off' images to remove that doesn't seem to make sense. Also, you can't subtract noise – subtracting two noisy images makes them more noisy. Averaging multiple images is the same as taking a longer exposure. It seems like what is fundamentally happening is the DL network is learning to produce an image that resembles a longer exposure image (with better SNR) than a noisy image. This isn't necessarily a bad thing, but how this process is presented doesn't make sense.

Response:

Noise in imaging systems can arise from various sources, such as shot noise, out-of-focus signals, autofluorescence, camera readout noise, and dark current noise. Traditional denoising methods face two key limitations. First, they typically target a single noise type or combine approaches based on individual noise characteristics. However, distinguishing specific noise types during actual imaging is challenging, and some noises remain uncharacterized, making it difficult to suppress all noise components simultaneously. Second, these methods primarily use information from surrounding pixels to perform mathematical operations on a target pixel. Because the signal and noise vary across different pixel locations, this computation alters how the signal intensity is distributed, which can interfere with subsequent resolution enhancement via RL deconvolution.

Essentially, 3S-denoising is a per-pixel denoising method that, unlike traditional approaches, removes most noise while preserving the spatial distribution of the original signal. A key advantage is that it requires no prior knowledge of the noise or background. Instead, for each pixel we classify noise into two modes based on temporal behavior: a random component that varies from frame to frame, and a fixed-pattern component that is stable across frames for that pixel. For instance, the dark-current noise has two parts: dark-current shot noise (random) and dark-current non-uniformity (a fixed spatial component). We reduce both random and fixed background/noise through optical modulation of photo-switchable fluorescent proteins. These proteins transition between ON/OFF states in response to specific wavelength modulation. Crucially, background/noise components remain unresponsive to this modulation. Both "ON" and "OFF" images contain background/noise with fixed and random patterns. We simultaneously extract signal and suppress background/noise through a computational process **combining subtraction and averaging operations**. The signal is extracted by subtracting the "OFF" state image from the "ON" state image, effectively removing or reducing both background and fixed-pattern noise. To further suppress random noise including Poisson noise, we perform temporal averaging across multiple frames of these on-minus-off difference images. The key advantage of this method lies in preserving the original signal distribution across pixels, enabling direct application in subsequent deconvolution processing.

We fully agree that subtracting two noisy images increases the overall noise level. However, this effect is effectively mitigated by frame averaging, as demonstrated in Fig. 1: averaging 50 ON-minus-OFF difference images yields a nearly 6× signal-to-noise ratio improvement, outperforming results obtained by averaging ON images alone (Fig. 1b, second vs. fifth column). While simple image averaging ON images or OFF images reduces random noise, it cannot remove fixed-pattern noise and background signals originating from the sample and imaging system (camera and optical path). To demonstrate this, we included images of non-fluorescent samples captured by the camera in simulation, which primarily contain these

background and fixed-pattern noise contributions. Our simulations reveal that these noise sources significantly degrade RL deconvolution performance, introducing artifacts and compromising resolution (Fig. 1b, second column). In contrast, 3Snet effectively suppresses fixed-pattern noise while minimizing RL-induced artifacts, as shown in Fig. 1b, fifth column. We believe the subtraction process in 3SNet plays a critical role in eliminating background and fixed-pattern noise. This effectiveness stems from the identical imaging conditions (e.g., excitation intensity, exposure time) used when capturing the "ON" and "OFF" images, ensuring consistent background and fixed-pattern contributions between the two frames. By subtracting the averaged "OFF" image from the averaged "ON" image, we effectively remove non-modulated background and noise components.

Fig. 1 | Workflow and validation of 3S-CLID on synthetic microtubule and cellular structure. **b**, Validation of 3S-CLID using synthetic microtubule. Top row: peak signal-to-noise ratio (pSNR) and structural similarity (SSIM), from left to right: noisy image, image averaging of ON image sequences, median of ON image sequences, median of the ON minus OFF image sequence, 3S-denoising result (Clear) and blurred ground truth. The color bar black to cyan means the relative intensity of images. Middle row: Data uncertainty results of top row indicated by the averaged standard deviation (STD). The color bar blue to red represents the STD value. Bottom row: deconvolution of noisy image, deconvolution of averaged ON image, deconvolution of median averaged ON image, deconvolution of median ON minus OFF image, 3S-CLID result and ground truth. The noisy images were created by acquiring a blank region of a fixed cell under WF illumination, and further injection of 5% Poisson noise and 5% Gaussian noise. The resulting images from various approaches were processed using Lanczos upsampling by a factor of 3, followed by RL deconvolution with a theoretically derived Bessel PSF, using 1000 iterations. **c**, Normalized intensity profiles along the red arrow heads in the left ground truth (GT) and deconvolved images.

We respectfully disagree with the position that averaging multiple images is equivalent to using a longer exposure. The critical distinction may lie in readout noise reduction: while both methods increase signal integration time, only frame averaging can reduce readout noise through temporal sampling (PMID: 31901080). This noise reduction is particularly important for RL deconvolution, as readout noise directly contributes to reconstruction artifacts. By employing frame averaging prior to deconvolution, we effectively minimize these noise-induced artifacts, yielding superior results compared to single long exposures.

We compared deconvolution results for a 3S-denoised image with those for a longer-exposure image of the same fixed-cell intracellular structure. Results showed that, compared to images

acquired with longer exposure times, stacking multiple short-exposure images yields stronger signal in high-frequency regions (indicated by arrows in panels a and c) and exhibits lower noise levels, as shown to the right of the dashed line in panels b and d.

We have added a description to the paper outlining the theoretical underpinnings of our noise treatment approach. (Line 119-145)

3. Comparing to a noise-free RL would be nice in the simulated images section.

Response:

Thank you for your valuable suggestion. We have now included a comparison with noise-free RL deconvolution in Fig. 1 of the revised manuscript.

4. The math in the Abstract is confusing because the numbers seem to suggest a factor of 4 improvement compared what is stated as 6.

Response:

Thank you for identifying this error. We have corrected it in the revised manuscript. According

to Fig. 5h, the averaged resolution was improved from 346 nm to 64 nm following 3Snet-CLID processing, improved 5.4 folds.

5. There are very minor grammatical issues.

Response:

Thank you for your comment. We have carefully reviewed the manuscript's grammar and made the necessary corrections.

POINT-BY-POINT RESPONSE TO REVIEWER'S COMMENTS

Reviewer #1 (Remarks to the Author):

The authors have responded sincerely to all comments with extensive additional experiments and detailed theoretical explanations. The technical quality of the work is high, and the comprehensive data disclosure demonstrates scientific integrity.

Response:

We appreciate your encouraging assessment of our work, as well as the time you devoted to reviewing this manuscript.

As pointed out in our initial Major Issues 1 and 2, the term "super-resolution" in the manuscript was unclear and misleading. Through the authors' earnest response to comments from all three reviewers, the essential nature of the method has been clarified: CLID is not an optical super-resolution microscopy technique but rather a computational super-resolution image processing method. This clarification of the paper's scope is highly valuable, and we commend the authors' sincere efforts in this regard. However, the title, abstract, and various sections of the main text still contain expressions and discussions that may mislead readers into confusing it with optical super-resolution microscopy. For example, phrases such as "surpasses the theoretical resolution threshold" and "recover lost details" suggest the measurement of new physical information beyond the diffraction limit, when in reality CLID performs computational estimation from existing measurements without hardware modifications.

We would like to suggest that rather than merely making superficial corrections to avoid misunderstandings, reorganizing the manuscript to clearly position it as a paper on "computational super-resolution image estimation/generation" would not only clarify its scope but also make it more compelling. The technical innovations—such as per-pixel denoising strategy, signal distribution preservation, and hybrid supervised/self-supervised learning—are currently scattered throughout the text. By integrating these elements and explicitly presenting the novelty and advantages as an image processing method, the paper's contribution would become clearer and more appealing to the broad readership of Nature Communications.

Response:

We greatly appreciate your valuable suggestions. We have now explicitly defined CLID as a computational super-resolution method, underscoring its advantages through key features such as the per-pixel denoising strategy, signal distribution preservation, and hybrid supervised/self-supervised learning. Additionally, we have revised the title, abstract, and discussion as you recommended to ensure that the manuscript consistently adopts the perspective of computational

super-resolution throughout.

Specific comments to the authors' revisions/rebuttals follow:

1. Major Issues 1 & 2: Contextualization and theoretical framework

The authors have provided extensive theoretical clarifications. We suggest this be reflected throughout the manuscript as discussed above.

Response:

Thank you for your valuable suggestion. To ensure the theoretical clarifications are thoroughly reflected in the manuscript, we have carefully revised the text throughout, including the Introduction, Methods, and Discussion sections, to integrate these points more seamlessly.

2. Major Issue 3: Image upsampling validation

We appreciate the comprehensive validation with systematic testing and complete data regeneration. Here, the manuscript reads that the synthetic ground truth has 65 nm pixel sampling (Lines 472-480). The validation showed 65 nm resolution achievement. The results can be interpreted in two ways: negatively, as circular validation where the information limit is predetermined; or positively, as evidence that CLID appropriately stops at the data's inherent limit without generating spurious information (avoiding hallucination). Given the importance of this point for interpreting CLID's capabilities, we suggest either more thorough discussion of this issue. More preferably, additional validation approaches if feasible.

Response:

We sincerely thank you for this exceptionally insightful comment, which raises a critical point regarding the interpretation of CLID's resolution capabilities. We fully agree that a thorough discussion is essential.

As you astutely noted, the validation results can be interpreted positively: CLID appropriately stops at the data's inherent resolution limit without generating spurious information, thereby avoiding "hallucination." We agree with this interpretation, and our existing data strongly support it. Below, we elaborate further and provide additional evidence to reinforce our conclusions:

1) The measured resolution is not predetermined by pixel sampling. The 65 nm value refers to the pixel sampling when $1\times$ upsampling is applied. In the original manuscript, we used $2\times$ upsampling, which resulted in a pixel size of 32.5 nm and a corresponding theoretical Nyquist limit of 65 nm. In our first revision, we switched to $3\times$ Lanczos upsampling, yielding a pixel size of 21.7 nm and a Nyquist limit of approximately 43.4 nm.

If the resolution were circularly predetermined by the upsampling step (the first, negative

interpretation), one would expect a measured resolution around 43 nm under $3\times$ upsampling. However, our actual measurements on various samples consistently yielded values larger than the camera's pixel sampling limit (~ 43.4 nm): Fig. 1c: ~ 65 nm (minimum distance between two beads); Fig. 2c & 2e: ~ 65 nm (close to designed distance between fluorophores); Fig. 1g: ~ 58 nm (cytoskeleton, measured by decorrelation analysis).

2) CLID faithfully reports the true structural dimensions. Quantitative validation on two reference samples confirms accuracy without significant artifact: i) DNA origami tiles with a designed 60-nm fluorophore spacing (Fig. 2e), yielded ~ 65 nm with CLID, matching STORM measurements. ii) Nup96 rings (Fig. 5a-c) gave an average diameter of ~ 103 nm, in close agreement with electron microscopy data. These results demonstrate that CLID resolves genuine geometry rather than introducing spurious features.

3) Direct evidence from dual-camera experiments. Following your excellent suggestion, we performed additional validation by imaging the same cellular structure using two cameras with different native pixel sizes (65 nm and 110 nm after optical magnification). The results (below) showed that with $2\times$ Lanczos upsampling, CLID achieved resolutions of 70 nm and 113 nm, very close to the respective pixel sampling limits. With $3\times$ upsampling, the resolutions values improved to ~ 60 nm and ~ 90 nm—still fundamentally limited by the Nyquist limit. This data provides direct evidence that CLID's resolving capability, like those of other RL deconvolution methods, is practically constrained by SNR and the pixel sampling, and that CLID algorithm correctly halts before reaching this boundary.

We have incorporated these points and the new dual-camera validation data into the revised manuscript. Specifically, we now cite "~60 nm" as the achieved resolution wherever appropriate and have added the following paragraph to the Discussion (Line 439-448): "The resolution of 3Snet-CLID, like that of other computational SR methods, is fundamentally bounded by the Nyquist limit of the pixel sampling and is practically constrained by the SNR. Under our current imaging conditions — 6.5 μm camera pixels, 100 \times objective, and 3 \times Lanczos upsampling — 3Snet-CLID achieves ~60 nm resolution on biological specimens without introducing appreciable artifacts. Fourier Ring Correlation (FRC) analysis of sector test pattern simulations (Supplementary Fig. 2) and of the same cellular structure imaged with two cameras whose native pixel differed (65 nm vs. 110 nm after optical magnification) (Supplementary Fig. 4) confirm that this gain arises from genuine high-frequency signal recovery rather than deconvolution artifacts. In principle, with higher SNR and finer pixel sampling, 3Snet-CLID can approach the corresponding Nyquist limit."

We believe these additions substantially strengthen the manuscript and provide a clearer, more compelling assessment of CLID's performance. We sincerely appreciate your insightful feedback, which has been instrumental in improving our work.

3. Major Issue 4: Comprehensive comparison analysis

We appreciate the addition of MED(ON-OFF) comparison and the detailed parameter specifications as requested. The results show CLID's advantages over alternative methods, which is encouraging. However, some numerical values in Fig. 1b, 1c are unexpected. For instance, MED(ON) shows STD of 15.6 versus AVG(ON)'s 5.14, and the performance gap between AVG(ON-OFF) and MED(ON-OFF) is larger than typically expected for Poisson noise. We suggest double-checking these values to ensure the comparison's reliability.

Response:

Thank you for this careful observation regarding the numerical values in Figures 1b and 1c. We have double-checked the calculations and can confirm the values are accurate.

We agree that the performance gap between AVG(ON-OFF) and MED(ON-OFF) is larger than would be expected for pure Poisson noise. We believe the explanation lies in the nature of our simulated data. The images were designed to incorporate not only Poisson noise but also other realistic noise types present in actual captured images. While MED(ON-OFF) is highly effective at suppressing Poisson noise, the superior performance of AVG(ON-OFF) in this case reflects its ability to more effectively mitigate these additional, non-Poisson noise components. This result underscores the robustness of the AVG(ON-OFF) method in a more realistic noise environment.

4. Major Issue 5: Demonstration of generalizability

We highly commend the authors' extensive efforts and sincere response. The comprehensive demonstration across diverse conditions (as summarized in Supplementary Table 2) successfully establishes CLID's excellent generalization capability, making its practical utility much clearer. This is a valuable contribution.

Response:

We are deeply grateful for your generous and encouraging comments. We are delighted that our efforts to demonstrate CLID's generalization capability have successfully conveyed its practical utility, and we thank you for recognizing the value of this contribution.

5. Minor Issues 1-5

We appreciate the authors' sincere and appropriate responses to all minor issues.

Response:

Thank you for this encouraging comment. We appreciate the acknowledgment and are glad our clarifications were satisfactory.

Reviewer #2 (Remarks to the Author):

Overall, the authors have satisfactorily addressed most of the concerns raised in the initial review, and the methodology is now much better explained. I appreciate the additional clarifications and improvements provided in this new version of the manuscript.

Response:

We are grateful to you for this encouraging feedback. We are delighted that the improved explanations and clarifications have satisfactorily addressed the concerns and enhanced the manuscript's methodology.

However, a few important points still deserve clarification or refinement:

1. It would be important to clearly state early in the manuscript (e.g., in the Introduction), that the proposed method is essentially two-dimensional and primarily suited for data with high optical sectioning, such as confocal spinning-disk, TIRF microscopy, or other flat samples. Since the approach relies on a 2D model of PSF blur, it does not account for 3D out-of-focus contributions, which inherently limits its applicability to thicker samples compared to standard 3D deblurring approaches. Explicitly stating this from the beginning would help readers understand the intended scope and limitations of the method.

Response:

Thank you for your suggestions. In response, we have now explicitly stated in the Introduction that CLID is two-dimensional, suitable for imaging samples with a depth of less than 10 μm and works best for confocal spinning-disk and TIRF microscopy imaging modes (Line 99-103).

2. I remain unconvinced by the choice of datasets, both simulated and experimental, used to demonstrate the generalizability of the approach, as they mainly feature 1D filamentous or network structures. I previously suggested, and still recommend, using simulated data that include objects of varying sizes (from micrometer to nanometer scale), such as sector test patterns. For the experimental part, data such as chromatin or lipid membrane staining would better represent more homogeneous 2D surfaces and could highlight that, once trained, the network can handle structures of diverse morphologies. Such examples would also better illustrate the robustness of the method against artefacts in homogeneous regions.

Response:

Thank you for the suggestion. Following the advice, we have incorporated simulated data containing structures of varying sizes (from micrometers to nanometers) to evaluate the recovery of fine detail resolution and to assess potential artifacts in homogeneous regions. Since the specific code for the sector test pattern referenced was unavailable, we generated a similar pattern covering a comparable range of spatial frequencies (shown below). The simulation results demonstrate that CLID not only greatly enhances image contrast but also resolves structures ranging from the micrometer to the nanometer scale, without introducing significant artifacts when imaging either homogeneous or heterogeneous structures.

For the experimental part, we would like to clarify that, based on evidence from techniques like label-free imaging and electron microscopy, both chromatin and lipid membrane staining structures are known to exhibit significant heterogeneity and are not perfectly homogeneous (PMID: 38306982; PMID: 27702891; PMID: 40645941; PMID: 28356571). Nevertheless, to directly address the valuable point about testing on diverse morphologies, we have applied CLID to image both chromatin and cell membranes. The results, presented below, successfully illustrate the method's capability on these biologically relevant structures. We have added both the simulation and experimental results to the manuscript (Lines 429-432).

- As artefact management is central to this work and to deconvolution in general, it would be useful to explicitly mention that, although artefacts are reduced compared to other algorithms (as illustrated in many comparative examples), residual texture artefacts may still occur, as visible in some figures (e.g., Fig. 4, central region, and Supplementary Video 1). A short comment on this point would guide readers to interpret results with appropriate caution. Such typical textures seem to occur in regions with low spatial details or dominated by out-of-focus signal.

Response:

Thank you for your valuable comment. We fully agree that open discussion of potential artifacts is essential. CLID raises pSNR from 8.2 to 47.5 and yields an SSIM of 0.98—performance that exceeds current denoising methods—yet it cannot remove every photon noise, and iterative deconvolution can still magnify residual artifacts.

Following your suggestion, we have inserted the following clarification into the revised manuscript: "Although 3Snet-CLID substantially improves image quality and suppresses artifacts relative to existing approaches (Figs. 2d, 3b, 3d, Extended Data Fig. 1), faint texture artifacts may occasionally persist, especially in low-SNR regions (e.g., the central area in Fig. 4a)." (Lines 437-439) and "CLID is a 2D SR technique and, like any algorithm that uses a 2D PSFs, remains susceptible to out-of-focus blur" (Lines 458-459)

We believe these statements give readers clear, balanced guidance on the capabilities and limitations of the technique.

- Typical artefacts are also visible in the Argolight slide imaging for all methods tested (Fig. 2d).

Even if 3Snet reconstructions outperform other tested methods, sometimes a single line is visible, and sometimes two, independently of the line-pair distances, which is unexpected and would require some explanations. It would be helpful to add a reference line above the figure showing the true pattern (e.g., are the patterns 1-2-1 or 2-2-2 lines?) and to clarify whether the line thickness remains constant across line-pair distances. Another puzzling point is why line-pairs larger than 120 nm are not properly resolved by any algorithms except 3Snet and sparse deconv, and why 3Snet seems to perform better on 60 nm pattern images compared to 210 nm ones. The poor performance of Noise2Self is somewhat surprising but can likely be explained by the fact that it was designed to denoise very noisy data. This suggests that evaluating the method's performance as a function of SNR, particularly in low-SNR conditions, would provide valuable insight into its limits.

Response:

Thank you for your valuable feedback. We have inserted a reference line above Figure 2d to indicate the expected line pattern (1-2-1). Because the artifacts appeared with every method in the original and first revision manuscript, we suspected a manufacturing defect in the original Argolight slide. We therefore repeated the acquisition on a newly purchased slide; the revised figure now shows these cleaner data, which fully address the anomalies you noted. Owing to space constraints, only the 30–210 nm data were shown in the main figure.

Thank you for highlighting the Noise2Self comparison. Our systematic tests show that

Noise2Self's pixel-subset strategy distorts the photon distribution; when the resulting image is passed to RL deconvolution this distortion produces severe artifacts and consistently low SSIM values, irrespective of SNR. Consequently, Noise2Self-deconvolution performs poorly even under high-SNR conditions, so the steep drop we observe is intrinsic to the method rather than a noise-level effect. In contrast, CLID preserves the original distribution and therefore retains a clear resolution advantage across the entire SNR range, including the lowest doses we tested.

- Finally, I remain cautious regarding the reported absolute resolution value of 65 nm (stated directly in the title). This could reflect algorithmic convergence effects (i.e., texture artefacts) rather than true structural resolution. For example, decorrelation analysis of deconvolved images such as those in Fig. 2d might yield similar or even smaller values, making it difficult to discriminate resolution improvement from artefact enhancement. I understand that it is not a trivial task, and including sector test patterns simulations covering a broad range of spatial frequencies could help clarify this point. Unless such analysis is provided, I would recommend moderating or qualifying this claim.

Response:

We appreciate your caution regarding the 65 nm value and share your concern about distinguishing genuine resolution from algorithmic artifacts. To address this, we now quantify resolution with Fourier-ring correlation (FRC), a metric specifically designed to separate

reproducible signal from noise-based artifacts (Nieuwenhuizen et al., Nat. Methods 2013).

1) Sector patterns simulations spanning a wide frequency range give an FRC resolution of ~ 55 nm after CLID and show no visible texture artifacts (SSIM = 0.89 versus ground truth), confirming that the gain comes from restored high-frequency signal, not noise amplification.

2) Direct evidence from dual-camera experiments. We performed additional validation by imaging the same cellular structure using two cameras with different native pixel sizes (65 nm and 110 nm after optical magnification). The results (below) showed that with $2\times$ Lanczos upsampling, CLID achieved resolutions of 70 nm and 113 nm, very close to the respective pixel sampling limits. With $3\times$ upsampling, the resolutions values improved to ~ 60 nm and ~ 90 nm—still fundamentally limited by the Nyquist limit. This data provides direct evidence that CLID's

resolving capability, like those of other RL deconvolution methods, is practically constrained by SNR and the pixel sampling, and that CLID algorithm correctly halts before reaching this boundary.

3) CLID faithfully reports the true structural dimensions. Quantitative validation on two reference samples confirms accuracy without significant artifact: i) DNA origami tiles with a designed 60-nm fluorophore spacing (Fig. 2e), yielded ~ 65 nm with CLID, matching STORM measurements. ii) Nup96 rings (Fig. 5a-c) gave an average diameter of ~ 103 nm, in close

agreement with electron microscopy data. These results demonstrate that CLID resolves genuine geometry rather than introducing spurious features.

In the original manuscript and the first revision, we reported the resolution of CLID as ~65 nm, based on the minimum resolvable distance measured between adjacent structures in standard samples (Fig. 1c, 2c, 2e). However, due to the limited number of pixels, a two-peak fitting could not be reliably performed; the measured “minimum resolvable distance” (65 nm) actually represents the distance between the pixel positions of two intensity maxima, which does not strictly equate to the actual distance between fluorophores after accounting for pixel discretization. Therefore, in this revision, we now use the FRC-derived resolution from biological samples to represent the resolution achievable by CLID. Furthermore, following your suggestion, we have qualified the reported resolution by stating that under our current imaging conditions (6.5 μm pixel camera, 100 \times objective, 3 \times Lanczos upsampling), CLID achieves a resolution of approximately 60 nm for biological samples without introducing significant artifacts. Accordingly, we have also removed the specific “65 nm” reference from the title.

We have incorporated these points and the new dual-camera validation data into the revised manuscript. Specifically, we have updated the description of CLID's achieved resolution to “~60 nm” throughout the manuscript where appropriate. And the Discussion has been expanded to include a clear paragraph stating (Line 439-448): “The resolution of 3Snet-CLID, like that of other computational SR methods, is fundamentally bounded by the Nyquist limit of the pixel sampling and is practically constrained by the SNR. Under our current imaging conditions — 6.5 μm camera pixels, 100 \times objective, and 3 \times Lanczos upsampling — 3Snet-CLID achieves ~60 nm resolution on biological specimens without introducing appreciable artifacts. Fourier Ring Correlation (FRC) analysis of sector test pattern simulations (Supplementary Fig. 2) and of the same cellular structure imaged with two cameras whose native pixel differed (65 nm vs. 110 nm after optical magnification) (Supplementary Fig. 4) confirm that this gain arises from genuine high-frequency signal recovery rather than deconvolution artifacts. In principle, with higher SNR and finer pixel sampling, 3Snet-CLID can approach the corresponding Nyquist limit.”

We believe these additions substantially strengthen the manuscript and provide a clearer, more convincing interpretation of CLID's performance. We are deeply grateful for your valuable feedback, which has been instrumental in improving our work.

Reviewer #3 (Remarks to the Author):

1. The authors have provided a thorough response to the reviewers. The comparison of the results of this method to deconvolution of a blurry, but noise free image are nearly identical (fig 2b). If this is truly the performance of the method, than this case should be made even more strongly

by the authors.

Response:

We thank you for highlighting this key observation. We agree that the near-identical performance between 3S-CLID applied to the noisy measurement and deconvolution applied to a noise-free blurred image (Fig. 1b, we believe you are referring to Fig. 1b) is an important result and should be emphasized more strongly in the manuscript.

Our CLID pipeline consists of (i) 3S denoising applied to the noisy image, followed by (ii) RL deconvolution applied to the denoised output. The main reason CLID approaches the “noise-free blurred” deconvolution baseline is that our 3S method is a single-pixel-based denoiser that can remove strong noise efficiently while faithfully preserving the per-pixel signal distribution (i.e., without introducing spatial hallucination or structural distortions). Consequently, the denoised image is a highly accurate estimate of the underlying blurred signal, enabling the subsequent RL step to behave almost the same as it would on a truly noise-free blurred input.

We have strengthened the manuscript accordingly by explicitly stating this result in the Results section around Fig. 1b and by adding the corresponding quantitative comparison. Specifically, for the example in Fig. 1b, the raw noisy image has $pSNR/SSIM = 8.2/0.15$; after 3S denoising the image reaches $47.5/0.98$; after applying RL (i.e., 3S-CLID) we obtain a super-resolved reconstruction of $24.5/0.93$, which is essentially indistinguishable from RL deconvolution of a blurry but noise-free image ($23.9/0.94$). This substantiates that CLID can deliver high-fidelity super-resolution under severe noise by first producing a denoised image that preserves the true signal distribution required for stable deconvolution.

We have added this part into the Results section: “Notably, Figure 1b highlights an important property of 3S-CLID: the 3S-CLID reconstruction obtained from a strongly noisy measurement is nearly identical to the reconstruction obtained by deconvolving a blurry but noise-free image. Starting from the raw noisy input ($pSNR/SSIM = 8.2/0.15$), 3S denoising step substantially suppresses noise while preserving the per-pixel signal distribution, resulting in a denoised estimate with $pSNR/SSIM = 47.5/0.98$. Applying RL deconvolution to this denoised image (i.e., 3S-CLID) yields a super-resolved reconstruction with $pSNR/SSIM = 24.5/0.93$, which closely matches the result of deconvolution applied to a blurry, noise-free image ($pSNR/SSIM = 23.9/0.94$). This near-equivalence indicates that 3S effectively recovers an accurate blurred signal estimate under severe noise, enabling the downstream deconvolution to operate in a regime comparable to the ideal noise-free case. Overall, 3S-CLID combines (i) efficient single-pixel-based denoising, (ii) faithful preservation of the signal distribution, and (iii) high-fidelity SR reconstruction.” (Lines 219-230)

2. A few critical issues remain - please explain exactly how the decorrelation analysis was done.

This should be by comparing FRC between two completely different data sets on the same sample, otherwise the high frequency info could just be a high-resolution, but wrong.

Response:

Thank you for your insightful comments. In our study, all SR images subjected to decorrelation analysis were first inspected visually at an appropriate iteration number to ensure no obvious texture artifacts were present before the analysis was performed. Consequently, the resolution values obtained from the decorrelation analysis correspond directly to the images shown in the figures.

We have cited the original method reference (Nature Methods, 2019, 16, 918–924) in the manuscript. In the revised version, we have also provided a detailed, step-by-step description of the decorrelation analysis procedure in the legend of Figure 1e (where it first appears), as follows: “We used the plugin ‘ImageDecorrelationAnalysis_plugin.jar’ to perform the decorrelation analysis. All SR images subjected to decorrelation analysis were first inspected visually at an appropriate iteration number to ensure no obvious texture artifacts were present before the analysis was performed.” (See Line 685-691)

In our revised manuscript (Line 444-448), we have performed additional validation by imaging the same cellular structure using two cameras with different pixel sizes (65 nm and 110 nm after

optical magnification). For each camera, we applied 3Snet denoising followed by RL deconvolution with both $2\times$ and $3\times$ Lanczos upsampling. At the appropriately chosen iteration numbers—where no obvious visual artifacts were present—FRC analysis was conducted on the independently acquired datasets. The results show that the resolutions determined by FRC and by decorrelation analysis are in close agreement under both upsampling conditions.

We believe this additional validation strengthens the methodological rigor of our work and addresses your concern regarding the interpretation of high-frequency information.

3. Please also clarify lambda in the loss model.

Response:

We thank the reviewer for pointing out the insufficient clarification of Lambda. We have added a comprehensive explanation in Methods (Line 511-518, highlighted in yellow): “ $\lambda > 0$ denotes the weighting factor between the self-supervised term and the supervised term in the loss function. A larger λ indicates a higher weight assigned to the supervised term. When $\lambda = 0$, the supervised term becomes zero and the DL model reduces to a self-supervised model; when $\lambda = \infty$, the self-supervised term becomes zero and the DL model reduces to a fully supervised model. To determine the optimal λ , we set λ to 0/1/2/4/8/16/ ∞ and trained the model on the same dataset for each setting. The dataset consists of two parts: an image set for training the denoising network (train) and an image set for evaluating the denoising performance (test). The λ used in the loss function is chosen as the value that yields the highest average pSNR on the denoised images in the test set.”

4. I share many of the concerns expressed by the other reviewers in the original reviews, in particular the requirement of the PSF to be known exactly as deconvolution goes much beyond the diffraction limit, otherwise artifacts can be produced. This should be discussed more strongly - although I see there is now a figure comparing the result when using mismatched PSFs.

Response:

We thank you for this important comment and agree that RL deconvolution is highly sensitive to PSF mismatch. When the PSF used in the forward model does not match the true system response, RL can over-fit the mismatch and introduce non-physical structures (e.g., ringing/haloing or spurious fine details), particularly when attempting to recover high spatial frequencies. We have therefore strengthened the manuscript text to emphasize this limitation explicitly and to clarify that accurate, condition-matched PSF estimation and conservative

iteration stopping are essential to avoid artifacts; we also refer the reader to the mismatch comparison now provided (Supplementary Fig. 6).

To ensure a matched PSF in our experiments, we used two complementary approaches under the same imaging configuration. First, we imaged sub-diffraction fluorescent beads on the same microscope and extracted the effective lateral/axial widths (e.g., FWHM) and sampling parameters; together with the known optical parameters (objective/NA, emission wavelength, pixel size), we generated a condition-matched PSF for RL deconvolution (Methods, revised). Second, we independently measured the PSF experimentally using recently reported single-chain ultrasmall fluorescent polymer dots (“suPdots”/polymerdots; <5 nm size and very high brightness), which behave as quasi-point emitters and enable direct PSF measurement under identical conditions (Yang et al., *Nature Photonics* 19, 1336–1344 (2025), doi:10.1038/s41566-025-01767-1). Importantly, RL deconvolution of the same denoised input image using (i) the bead-derived, condition-matched PSF and (ii) the experimentally measured polydot PSF

produced nearly indistinguishable results (new Supplementary Fig. 6), supporting that our PSF construction is accurate enough for the conclusions drawn and that the observed structures are not driven by an arbitrary PSF choice.

Finally, to minimize over-iteration artifacts, we applied a conservative stopping criterion and typically used appropriate RL iterations, terminating iterations before any visible onset of ringing/haloing or other obvious non-physical features. We now state this explicitly in the Methods and point to Supplementary Fig. 6 for the PSF-validation comparison and the robustness assessment.

A POINT-BY-POINT RESPONSE TO THE REVIEWERS' COMMENTS

Reviewer #1 (Remarks to the Author):

General Assessment

The revised manuscript represents a substantial improvement. The explicit positioning as "computational super-resolution," the removal of the specific resolution figure from the title, and the honest discussion of limitations (2D constraints, PSF sensitivity, residual artifacts, out-of-focus susceptibility) set an appropriate standard for reporting computational SR methods. Aside from one remaining concern discussed below, the revisions adequately address the reviewers' comments.

Response:

Thank you very much for your thorough and constructive feedback throughout the review process. We are delighted to hear that the revised manuscript represents a substantial improvement and that the explicit positioning as "computational super-resolution," along with our honest discussion of limitations, meets the appropriate standards for reporting computational SR methods.

The authors are particularly commended for conducting additional dual-camera validation experiments—an effort that goes beyond mere textual revision.

Response:

We are particularly grateful for your recognition of our additional dual-camera validation experiments. Your acknowledgment that this effort goes beyond mere textual revision is deeply appreciated, as we indeed invested significant time and resources to strengthen the experimental validation of our method.

Remaining Concern

The use of decorrelation analysis for resolution estimation is understandable and not inappropriate per se. However, given that the authors have now explicitly framed this work as computational super-resolution, the limitations of this metric when applied to computational SR outputs should be clearly discussed.

Decorrelation analysis confirms the presence of high-frequency content but does not independently verify its accuracy. Hou et al. (eLight 2024) demonstrated that deconvolved images can show higher resolution by decorrelation analysis "largely due to simple amplification of the high-frequency part" while lacking fidelity. The 2020 Addendum to the original method (Descloux et al., Nature Methods) also acknowledges that image generation processes can influence resolution estimates.

Response:

We thank you for this important point. We agree that, when a method is framed as computational super-resolution, a single-image resolution metric such as decorrelation analysis should be interpreted cautiously, because it primarily confirms the presence of high-frequency content rather than independently proving its fidelity.

Following your suggestion, we revised the manuscript to more clearly discuss these limitations in the context of computational SR (Line 443-453, highlighted):

In the current study, we primarily employ decorrelation analysis to estimate image resolution; however, we acknowledge that this metric has limitations for computational super-resolution outputs. The 2020 Addendum to the original method notes that resolution estimates can be influenced by image-generation or post-processing steps (Descloux et al., Nature Methods, 2020), and Hou et al. (eLight, 2024) further demonstrated that deconvolution may yield artificially elevated decorrelation-based resolution largely through high-frequency amplification without a corresponding improvement in fidelity. Thus, decorrelation analysis confirms the presence of high-frequency content but does not independently verify its accuracy. To mitigate this risk, all SR images in our study were visually inspected at optimized iteration numbers to avoid artifacts prior to decorrelation analysis; under these conditions, the resulting resolution estimates closely agreed with FRC-based validation (Supplementary Fig. 4).

Suggestion

Part of this concern can be addressed using the authors' own dual-camera dataset. Cross-FRC analysis between the CLID-processed outputs from the two cameras is strongly recommended. Because the two cameras have different pixel sizes, algorithmically generated high-frequency artifacts would be pixel-grid-dependent and thus uncorrelated between the two outputs, whereas genuine structural information would correlate. This provides more rigorous validation than independent decorrelation analysis of each output and requires only re-analysis of existing data. If the authors' claims are correct, cross-FRC should show correlation up to ~90 nm, corresponding to the resolution of the lower-resolution camera output. This would provide independent confirmation that structural information in this frequency range is consistently recovered. The ~60 nm claim, however, cannot be validated with this dataset. If the authors wish to further validate higher frequencies, a dual-camera configuration with a smaller pixel size (e.g., using a 2× relay lens to achieve ~55 nm pixels) would be one possible approach.

Response:

We thank you for the suggestion to perform cross-FRC between the CLID outputs from the two cameras. We agree with the underlying intent (to test whether high-frequency information is consistently recovered across independent measurements). However, we believe that cross-FRC is not a rigorous or interpretable validation in our current dual-camera configuration, because the two cameras differ substantially in pixel pitch (65 μm vs 110 μm) and consequently produce images with different sampling grids and different fields of view. Standard two-image FRC assumes that the two images represent the same region of interest on a common sampling grid, with independent noise realizations (Koho, S., et al. Nature Communications 10, 3103 (2019).). In our case, to compute cross-FRC one would first need (i) geometric registration (translation/rotation and potentially distortion correction due to different effective magnification and FOV), and (ii) resampling of one image onto the other's pixel grid (or onto a third common grid). Both steps are known to affect the FRC frequency axis and correlation values because FRC computed from discretized images depends on the sampling step/pixel size and the numerical handling of the discrete Fourier domain.

Concretely, Koho et al. demonstrate that if one plots FRC curves while incorrectly assuming the

same pixel size, the curves show an apparent frequency shift, and they explicitly state that the calibration is needed to correct for different pixel pitch. This is directly relevant here: any cross-FRC between two differently sampled outputs is not “plug-and-play” and requires a careful calibration/transform pipeline. Moreover, Pham et al. (closed-form FRC analysis) emphasize that discrete FRC depends on experimental/numerical parameters such as pixel size, and that the accuracy of discrete FRC is tied to the spatial sampling step and interpolation/aliasing behavior. Importantly, after resampling and registration, the resulting cross-FRC would no longer measure the agreement of the native CLID outputs from each camera, but rather the agreement between one native output and a geometrically transformed/interpolated version of the other. In such a situation, cross-FRC becomes strongly confounded by (a) interpolation-induced correlations/spectral smoothing, and (b) residual registration errors, both of which can dominate the high-frequency correlation decay and lead to under- or over-estimation that is not attributable to true optical/super-resolved structural information.

For these reasons, in our view the dual-camera dataset is well-suited for independent within-camera validations (as we already performed: decorrelation analysis and FRC validation per camera, Supplementary Fig. 4), but it is not ideally suited for cross-FRC as a decisive fidelity test given the large pixel-pitch mismatch and different FOV requiring nontrivial resampling/registration. Finally, we agree with the reviewer that, if one aims to validate frequencies beyond the lower-sampling channel, an improved dual-camera configuration with a smaller effective pixel size (e.g., using a relay to reach ~55 nm sampling at the specimen plane) would be a more appropriate experimental design, because it would reduce the need for aggressive resampling and extend the interpretable frequency range.

Recommendation Regarding Resolution Claims

Given these validation limitations, stating a specific resolution figure in prominent locations is potentially misleading. The authors' decision to remove the specific number from the title reflects appropriate caution and is commendable. However, the claim of “~60 nm resolution” still appears in the Abstract. Consistent with the reasoning behind the title revision, I recommend removing this figure from the Abstract.

Response:

Thank you for this valuable suggestion. We have removed the claim of “~60 nm resolution” from the Abstract to maintain consistency with the title revision.

Reviewer #1 (Remarks on code availability):

The repository was not accessible at the time of review.

Response:

Thank you for bringing this to our attention. We have corrected the link, and the repository is now fully accessible.

Reviewer #2 (Remarks to the Author):

After carefully reviewing the final revised version of the manuscript, I am pleased to note that the authors have thoroughly addressed all the concerns raised during the previous rounds of review.

The revisions have substantially strengthened the manuscript, both in terms of methodological clarity and in the discussion of limitations and validation. I appreciate the authors' careful and constructive responses throughout the review process.

Response:

Thank you very much for your thorough and constructive feedback throughout the review process. We greatly appreciate your time and effort in carefully evaluating our manuscript across multiple rounds of revision.

We are delighted to hear that the revisions have successfully addressed your concerns and that the manuscript has been strengthened in terms of methodological clarity as well as the discussion of limitations and validation. Your insightful comments and suggestions have been invaluable in helping us improve the quality of our work.

Before final publication, I would kindly ask the authors to verify that the newly added supplementary figures mentioned in the point-by-point response to the reviewers are properly included in the merged final document, as they do not seem to appear in the current. In addition, it would be beneficial to include zoomed-in views of selected images to better visualize the reported improvements as well as potential artefact-prone regions. Such magnified views would help readers more clearly appreciate both the gains and the limitations of the approach.

Response:

We have verified that the newly added supplementary figures mentioned in the point-by-point response to the reviewers are properly included in the merged final document. And as suggested, we also included zoomed-in views of selected images to better visualize the reported improvements as well as potential artefact-prone regions.

Reviewer #3 (Remarks to the Author):

The authors have satisfactorily responded to my criticisms.

Response:

We greatly appreciate your thorough review and constructive feedback throughout the revision process. We are pleased that our responses and revisions have satisfactorily addressed your criticisms. Thank you for your valuable time and effort in helping us improve the quality of our manuscript.